# Timing is Everything: Learning to Act Selectively with Costly Actions and Budgetary Constraints

## Abstract

Many real-world settings involve costs for performing actions; transaction costs in financial systems and fuel costs being common examples. In these settings, performing actions at each time step quickly accumulates costs leading to vastly suboptimal outcomes. Additionally, repeatedly acting produces wear and tear and ultimately, damage. Determining *when to act* is crucial for achieving successful outcomes and yet, the challenge of efficiently *learning* to behave optimally when actions incur minimally bounded costs remains unresolved. In this paper, we introduce a reinforcement learning (RL) framework named **L**earnable **I**mpulse **C**ontrol **R**einforcement **A**lgorithm (LICRA), for learning to optimally select both when to act and which actions to take when actions incur costs. At the core of LICRA is a nested structure that combines RL and a form of policy known as *impulse control* which learns to maximise objectives when actions incur costs. We prove that LICRA, which seamlessly adopts any RL method, converges to policies that optimally select when to perform actions and their optimal magnitudes. We then augment LICRA to handle problems in which the agent can perform at most $k < \infty$ actions and more generally, faces a budget constraint. We show LICRA learns the optimal value function and ensures budget constraints are satisfied almost surely. We demonstrate empirically LICRA's superior performance against benchmark RL methods in OpenAI gym's *Lunar Lander* and in *Highway* environments and a variant of the Merton portfolio problem within finance.

## 1 Introduction

There are many settings in which agents incur costs each time they perform an action. Transaction costs in financial settings [19], fuel expenditure [32], toxicity as a side effect of controlling bacteria [29] and physical damage produced by repeated action that produces wear and tear are just a few among many examples [13]. In these settings, performing actions at each time step is vastly suboptimal since acting in this way results in prohibitively high costs and undermines the service life of machinery. Minimising wear and tear is an essential attribute to safeguard against failures that can result in catastrophic losses [13].

Reinforcement learning (RL) is a framework that enables autonomous agents to learn complex behaviours from interactions with the environment [30, 11]. Within the standard RL paradigm, determining optimal actions involves making a selection from among many (possibly infinite) actions; a procedure that must be performed at each time-step as the agent decides on an action. In unknown settings, the agent cannot immediately exploit any topological structure of the action set (if any exists). Consequently, learning *not to take* an action i.e performing a zero or *null action*, involves expensive optimisation procedures over the entire action set. Since this must be done at each state, this process is vastly inefficient for learning optimal policies when the agent incurs costs for acting.

In this paper, we tackle this problem by developing an RL framework for finding both an optimal criterion to determine whether or not to execute actions as well as learning optimal actions. A key component of our framework is a novel combination of RL with a form of policy known as *impulse*

*control* [22, 19]. This enables the agent to determine the appropriate points to perform an action as well as the optimal action itself. Despite its fundamental importance as a tool for tackling decision problems with costly actions, presently, the use of impulse control within learning contexts (and unknown environments) is unaddressed.

We present an RL impulse control framework called LICRA, which, to our knowledge, is the first learning framework for impulse control. To enable learning optimal impulse control policies in unknown environments, we devise a framework that consists of separate RL components for learning when to act and how to act optimally. The resulting framework is a structured two-part learning process which differs from current RL protocols. In LICRA, at each time step, the agent firstly makes a decision whether to act or not leading to a binary decision space $\{0, 1\}$ (we later show that this is determined by evaluating an easy-to-evaluate criterion which has the value function as its input). The second decision part determines the best action to take. This generates a subdivision of the state space into two regions; one in which the agent performs actions and another in which it does not act at all. This is extremely useful since the agent quickly determines the set of states to *not take* actions while performing actions only at the subset of states where actions are to be executed.

We then establish theory that ensures convergence of LICRA to an optimal policy for such settings. To do this, we give a series of results namely:

**i)** We establish a dynamic programming principle (DPP) for impulse control and show that the optimal value function can be obtained as a limit of a value iterative procedure (Theorem 1) which lays the foundation for an RL approach to impulse control.

**ii)** We extend result i) to a new variant of Q learning which enables the impulse control problem to be solved using our RL method (Theorem 2).

**iii)** We characterise the optimal conditions for performing an action which we reveal to be a simple 'obstacle condition' involving the agent's value function (Prop. 1). Using this, the agent can quickly determine whether or not it should act and if so, then learn what the optimal action is.

**iv)** We then extend the result i) to (linear) function approximators enabling the value function to be parameterised (Theorem 3).

**iv)** In Sec. 6, we extend LICRA to include budgetary constraints so that each action draws from a fixed budget which the agent must stay within. Analogous to the development of i), we establish another DPP from which we derive a Q-learning variant for tackling impulse control with budgetary constraints (Theorem 4). A particular case of a budget constraint is when the number of actions the agent can take over the horizon is capped.

Lastly, we perform a set of experiments to validate our theory within the *Highway* driving simulator and OpenAI's *LunarLander* [7].

LICRA confers a series of advantages. As we demonstrate in our experiments, LICRA learns to compute the optimal problems in which the agent faces costs for acting in an efficient way which outperforms leading RL baselines. Second, as demonstrated in Sec. 6 LICRA handles settings in which the agent has a cap the total number of actions it is allowed to execute and more generally, generic budgetary constraints. LICRA is able to accommodate any RL base algorithm unlike various RL methods designed to handle budgetary constraints.

## 2  Related Work

In continuous-time optimal control theory [24], problems in which the agent faces a cost for each action are tackled with a form of policy known as *impulse control* [22, 19, 2]. In impulse control frameworks, the dynamics of the system are modified through a sequence of discrete actions or bursts chosen at times that the agent chooses to apply the control policy. This distinguishes impulse control models from classical decision methods in which an agent takes actions at each time step while being tasked with the decision of only which action to take. Impulse control models represent appropriate modelling frameworks for financial environments with transaction costs, liquidity risks and economic environments in which players face fixed adjustment costs (e.g. *menu costs*) [16, 20].

The current setting is intimately related to the *optimal stopping problem* which widely occurs in finance, economics and computer science [23, 31]. In the optimal stopping problem, the task is to determine a criterion that determines when to arrest the system and receive a terminal reward. In this case, standard RL methods are unsuitable since they require an expensive sweep (through the set of states) to determine the optimal point to arrest the system. The current problem can be viewed

as an augmented problem of optimal stopping since the agent must now determine both a sequence of points to perform an action or *intervene* and their optimal magnitudes — only acting when the cost of action is justified [25]. Adapting RL to tackle optimal stopping problems has been widely studied [31, 4, 9] and applied to a variety of real-world settings within finance [12], radiation therapy [1] and network operating systems [3]. Our work serves as a natural extenstion to RL approaches to optimal stopping to the case in which the agent must decide at which points to take many actions. As with optimal stopping, standard RL methods cannot efficiently tackle this problem since determining whether to perform a 0 action requires a costly sweep through the action space at every state [31]. In [26] the authors introduce "sparse action" with a similar motivation as impulse control. However, the authors treat only the discrete action space case. The authors in [26] do not discuss a broader theoretical framework of dealing with "sparse actions", and develop purely algorithmic solutions. Additionally, unlike the approach taken in [26], the problem setting we consider is one in which the agent faces a cost for each action - the produces a need for the agent to be selective about where it performs actions (but does not necessarily constrain the magnitude or choice of those actions).

## 3   Preliminaries

**Reinforcement Learning (RL).** In RL, an agent sequentially selects actions to maximise its expected returns. The underlying problem is typically formalised as an MDP $\langle \mathcal{S}, \mathcal{A}, P, R, \gamma \rangle$ where $\mathcal{S} \subset \mathbb{R}^p$ is the set of states, $\mathcal{A} \subset \mathbb{R}^k$ is the set of actions, $P : \mathcal{S} \times \mathcal{A} \times \mathcal{S} \to [0, 1]$ is a transition probability function describing the system's dynamics, $R : \mathcal{S} \times \mathcal{A} \to \mathbb{R}$ is the reward function measuring the agent's performance and the factor $\gamma \in [0, 1)$ specifies the degree to which the agent's rewards are discounted over time [30]. At time $t \in 0, 1, \ldots$, the system is in state $s_t \in \mathcal{S}$ and the agent must choose an action $a_t \in \mathcal{A}$ which transitions the system to a new state $s_{t+1} \sim P(\cdot|s_t, a_t)$ and produces a reward $R(s_t, a_t)$. A policy $\pi : \mathcal{S} \times \mathcal{A} \to [0, 1]$ is a probability distribution over state-action pairs where $\pi(a|s)$ represents the probability of selecting action $a \in \mathcal{A}$ in state $s \in \mathcal{S}$. The goal of an RL agent is to find a policy $\hat{\pi} \in \Pi$ that maximises its expected returns given by the value function: $v^\pi(s) = \mathbb{E}[\sum_{t=0}^\infty \gamma^t R(s_t, a_t)|a_t \sim \pi(\cdot|s_t), s_0 = s]$ where $\Pi$ is the agent's policy set. The action value function is given by $Q(s, a) = \mathbb{E}[\sum_{t=0}^\infty R(s_t, a_t)|a_0 = a, s_0 = s]$.

We consider a setting in which the agent faces at least some minimal cost for each action it performs. With this, the agent's task is to maximise:

$$v^\pi(s) = \mathbb{E}\left[\sum_{t=0}^\infty \gamma^t \left\{ \mathcal{R}(s_t, a_t) - \mathcal{C}(s_t, a_t) \right\} \Big| s_0 = s \right], \tag{1}$$

where for any state $s \in \mathcal{S}$ and any action $a \in \mathcal{A}$, the functions $\mathcal{R}$ and $\mathcal{C}$ are given by $\mathcal{R}(s, a) = R(s, a)\mathbf{1}_{a \in \mathcal{A}} + R(s, 0)(1 - \mathbf{1}_{a \in \mathcal{A}})$ where $\mathbf{1}_{a \in \mathcal{A}}$ is the indicator function which is 1 when $a \in \mathcal{A}$ and 0 otherwise and $\mathcal{C}(s, a) := c(s, a)\mathbf{1}_{a \in \mathcal{A}}$ where $c : \mathcal{S} \times \mathcal{A} \to \mathbb{R}$ is a minimally bounded (cost) function[1] that introduces a cost each time the agent performs an action. Examples of the cost function is a quasi-linear function of the form $c(s_t, a_t) = \kappa + f(a_t)$ where $f : \mathcal{A} \to \mathbb{R}_{>0}$ and $\kappa$ is a positive real-valued constant. Since acting at each time step would incur prohibitively high costs, the agent must be selective when to perform an action. Therefore, in this setting, the agent's problem is augmented to learning both an optimal policy for its actions and, learning at which states to apply its action policy.

**Example: Merton Portfolio Problem with Transaction Costs [10].** An investor performs a series of costly portfolio adjustments by buying and selling amounts of different assets within their portfolio. Each investment incurs a fixed minimal cost (also known as *transaction costs*) which is deducted from the investor's available cash-flow. The investor's aim is to maximise their total wealth (the value of the sum of their assets) at some time horizon by adjusting their portfolio of investments. Problems of this kind, portfolio investment problems are of fundamental importance within finance [18].
**Example 2.** An autonomous vehicle must perform a series of actions to perform a task. Each action draws from its fuel budget. In order to complete its task successfully, during the task, the vehicle must ensure it maintains an available supply.

---

[1]I.e. a function which is bounded below by a positive constant.

## 4 The LICRA Framework

In RL, the agent's problem involves learning to act at *every* state including those in which actions do not significantly impact on its total return. While we can add a zero action to the action set $\mathcal{A}$ and apply standard methods, we argue that this may not be the best solution in many situations. We argue the optimal policy has the following form:

$$\widetilde{\pi}(\cdot|s) = \begin{cases} a_t & s \in \mathcal{S}_I, \\ 0 & s \notin \mathcal{S}_I, \end{cases} \tag{2}$$

which implies that we simplify policy learning by determining the set $\mathcal{S}_I$ first — the set where we actually need to learn the policy.

We now introduce a learning method for producing impulse controls. This enables the agent to learn to select states to perform actions. Therefore, now agent is tasked with learning to act at states that are most important for maximising its total return given the presence of the cost for each action. Now at each state the agent first makes a *binary decision* to decide to perform an action.

Our framework, LICRA consists of two core components: firstly a RL process $\mathfrak{g} : \mathcal{S} \times \{0, 1\} \to [0, 1]$ and a second RL process $\pi : \mathcal{S} \times \mathcal{A} \to [0, 1]$. The role of $\mathfrak{g}$ is to determine whether or not an action is to be performed by the policy $\pi$ at a given state $s$. If activated, the policy $\pi$ determines the action to be selected. Prior to decisions being made, the policy $\pi$ communicates to $\mathfrak{g}$ the action it would take. An important feature of our LICRA is the *sequential decision process*. In LICRA, the policy $\pi$ first proposes an action $a \in \mathcal{A}$ which is observed by the policy $\mathfrak{g}$. Therefore, the role of $\mathfrak{g}$ is to prevent actions for which the change in expected future rewards does not exceed the costs incurred for taking such actions. By isolating a decision policy over whether an action should be taken or not, the impulse controls mechanism results in a framework in which the problem facing the agent has a markedly reduced decision space (in comparison to a standard RL method). Crucially, the agent must compute optimal actions at only a subset of states which are chosen by the policy $\mathfrak{g}$. Below is the pseudocode for LICRA, we provide full details of the code in Sec. 9 of the Appendix.

---

**Algorithm 1:** **L**earnable **I**mpulse **C**ontrol **R**einforcement **A**lgorithm (LICRA)

1: **Input:** Stepsize $\alpha$, batch size $B$, episodes $K$, steps per episode $T$, mini-epochs $e$
2: **Initialise:** Policy network (acting) $\pi$, Policy network (switching) $\mathfrak{g}$,
    Critic network (acting )$V_\pi$,Critic network (switching )$V_\mathfrak{g}$
3: Given reward objective function, $R$, initialise Rollout Buffers $\mathcal{B}_\pi, \mathcal{B}_\mathfrak{g}$ (use Replay Buffer for SAC)
4: **for** $N_{episodes}$ **do**
5:     Reset state $s_0$, Reset Rollout Buffers $\mathcal{B}_\pi, \mathcal{B}_\mathfrak{g}$
6:     **for** $t = 0, 1, \ldots$ **do**
7:         Sample $a_t \sim \pi(\cdot|s_t)$
8:         Sample $g_t \sim \mathfrak{g}(\cdot|s_t)$
9:         **if** $g_t = 0$ **then**
10:             Apply $a_t$ so $s_{t+1} \sim P(\cdot|a_t, s_t)$,
11:             Receive rewards $r_t = \mathcal{R}(s_t, a_t)$
12:             Store $(s_t, a_t, s_{t+1}, r_t)$ in $\mathcal{B}_\pi$
13:         **else**
14:             Apply the null action so $s_{t+1} \sim P(\cdot|0, s_t)$,
15:             Receive rewards $r_t = \mathcal{R}(s_t, 0)$.
16:         **end if**
17:         Store $(s_t, g_t, s_{t+1}, r_t)$ in $\mathcal{B}_\mathfrak{g}$
18:     **end for**
19:     **// Learn the individual policies**
20:     Update policy $\pi$ and critic $V_\pi$ networks using $\mathfrak{B}_\pi$
21:     Update policy $\mathfrak{g}$ and critic $V_\mathfrak{g}$ networks using $\mathfrak{B}_\mathfrak{g}$
22: **end for**

---

While we consider now two policies $\pi, \mathfrak{g}$, the cardinality of the action space does not change. In the discrete case the cardinality is still $|\mathcal{A}| + 1$, where $|\mathcal{A}|$ is cardinality of the action space for policy $\pi$.

Although action space cardinality does not change there are still benefits of using impulse control mechanism. This mechanism forces the agent to first determine the set of states to perform actions *only then* determine the optimal actions at these states. An important fact to note is that the decision space for the determining whether or not to execute an action is $\mathcal{S} \times \{0, 1\}$ i.e at each state it makes a binary decision. Consequently, the learning process for aspect is much quicker than a policy which must optimise over a decision space which is $|\mathcal{S}||\mathcal{A}|$ (choosing an action from its action space at every state). This results in the agent rapidly learning which states to focus on to learn which actions to perform. In the case of $\pi$ with a continuous action space again the impulse control mechanism does not change the cardinality of the action space. However, if the set $\mathcal{S}/\mathcal{S}_I$, where the optimal policy chooses $0$, is large enough, then again it can be more efficient to learn $\mathfrak{g}$ first and only then learn $\pi$ (we later validate this claim empirically, see Sec. 11.2),

In Sec. 5, we prove the convergence properties of LICRA. LICRA consists of two independent procedures: a learning process for the policy $\pi$ and simultaneously, a learning process for the impulse policy $\mathfrak{g}$ which determines at which states to perform an action. In our implementation, we used proximal policy optimisation (PPO) [27] for the policy $\pi$ and for the impulse policy $\mathfrak{g}$, whose action set consists of two actions (intervene or do not intervene) we used a soft actor critic (SAC) process [14] LICRA is a plug & play framework which enables these RL components to be replaced with any RL algorithm of choice.

# 5 Convergence and Optimality of LICRA

A key aspect of our framework is the presence of two RL processes that make decisions in a sequential order. In order to determine when to act the policy $\mathfrak{g}$ must learn the states to allow the policy $\pi$ to perform an action which the policy $\pi$ must learn to select optimal actions whenever it is allowed to execute an action.

In this section, we prove that LICRA converges to an optimal solution of the system. Central to LICRA is a Q-learning type method which is adapted to handle RL settings in which the agent must also learn when to act. We then extend the result to allow for (linear) function approximators. We provide a result that shows the optimal intervention times are characterised by an 'obstacle condition' which can be evaluated online therefore allowing the $\mathfrak{g}$.

Given a function $Q : \mathcal{S} \times \mathcal{A} \to \mathbb{R}$, $\forall \pi, \pi' \in \Pi$ and $\forall \mathfrak{g}, \mathfrak{g}'$, $\forall s_{\tau_k} \in \mathcal{S}$, we define the intervention operator $\mathcal{M}^{\pi,\mathfrak{g}}$ by $\mathcal{M}^{\pi,\mathfrak{g}} Q^{\pi',\mathfrak{g}'}(s_{\tau_k}, a_{\tau_k}) := \mathcal{R}(s_{\tau_k}, a_{\tau_k}) - c(s_{\tau_k}, a_{\tau_k}) + \gamma \sum_{s' \in \mathcal{S}} P(s'; a_{\tau_k}, s) v^{\pi',\mathfrak{g}'}(s') \Big| a_{\tau_k} \sim \pi(\cdot|s_{\tau_k})$, where $\tau_k$ is an intervention time.

The interpretation of $\mathcal{M}$ is the following: suppose that the agent is using the policy $\pi$ and at time $\tau_k$ the system is at a state $s_{\tau_k}$ and the agent performs an action $a_{\tau_k} \sim \pi(\cdot|s_{\tau_k})$. A cost of $c(s_{\tau_k}, a_{\tau_k})$ is then incurred by the agent and the system transitions to $s' \sim P(\cdot; a_{\tau_k}, s_{\tau_k})$. Lastly, recall $v^{\pi,\mathfrak{g}}$ is the agent value function under the policy pair $(\pi, g)$. Therefore, the quantity $\mathcal{M}Q^{\pi,\mathfrak{g}}$ measures the expected future stream of rewards after an immediate intervention minus the cost of intervention. This object plays a crucial role in the LICRA framework which as we later discuss, exploits the cost structure of the problem to determine when the agent should perform an intervention.

Given a function $v^{\pi,\mathfrak{g}} : \mathcal{S} \to \mathbb{R}$, we define the Bellman operator $T$, by:

$$Tv^{\pi,\mathfrak{g}}(s) := \max \Big\{ \mathcal{M}^{\pi,\mathfrak{g}} Q^{\pi,\mathfrak{g}}(s, a), \mathcal{R}(s, 0) + \gamma \sum_{s' \in \mathcal{S}} P(s'; 0, s) v^{\pi,\mathfrak{g}}(s') \Big\}, \qquad \forall s \in \mathcal{S}. \qquad (3)$$

The Bellman operator captures the nested sequential structure of the LICRA algorithm. In particular, the structure in (3) consists of an inner structure which consists of two terms: the first term is the expected future return given an action is taken at the current state under the policy $\pi$. The second term is the expected future return given no action is taken at the current state. Lastly, the outer structure is an optimisation which compares the expected return of the two possibilities and selects the maximum.

Our first result proves $T$ is a contraction operator in particular, the following bound holds:

**Lemma 1** *The Bellman operator $T$ is a contraction, that is the following bound holds:*

$$\|Tv - Tv'\| \le \gamma \|v - v'\|,$$

where $v, v'$ are elements of a finite normed vector space. We can now state our first main result:

**Theorem 1** *Given any $v^{\pi, \mathfrak{g}} : \mathcal{S} \times \mathcal{A} \to \mathbb{R}$, the optimal value function is given by $\lim_{k \to \infty} T^k v^{\pi, \mathfrak{g}} = \max_{\hat{\pi}, \hat{g} \in \Pi} v^{\hat{\pi}, \hat{g}} = v^{\pi^\star, g^\star}$ where $(\pi^\star, g^\star)$ is the optimal policy pair.*

The result of Theorem 1 enables the solution to the agent's impulse control problem to be determined using a value iteration procedure. Moreover, Theorem 1 enables a Q-learning approach [6] for finding the solution to the agent's problem.

**Theorem 2** *Consider the following Q learning variant:*

$$Q_{t+1}(s_t, a_t) = Q_t(s_t, a_t)$$
$$+ \alpha_t(s_t, a_t) \left[ \max \left\{ \mathcal{M}^{\pi, \mathfrak{g}} Q_t(s_t, a_t), \mathcal{R}(s_t, 0) + \gamma \max_{a' \in \mathcal{A}} Q_t(s_{t+1}, a') \right\} - Q_t(s_t, a_t) \right], \quad (4)$$

*then $Q_t$ converges to $Q^\star$ with probability 1, where $s_t, s_{t+1} \in \mathcal{S}$ and $a_t \in \mathcal{A}$.*

We now extend the result to (linear) function approximators:

**Theorem 3** *Given a set of linearly independent basis functions $\Phi = \{\phi_1, \ldots, \phi_p\}$ with $\phi_k \in L_2, \forall k$. LICRA converges to a limit point $r^\star \in \mathbb{R}^p$ which is the unique solution to $\Pi \mathfrak{F}(\Phi r^\star) = \Phi r^\star$ where $\mathfrak{F} v := \mathcal{R} + \gamma P \max\{\mathcal{M} v, v\}$. Moreover, $r^\star$ satisfies: $\|\Phi r^\star - Q^\star\| \leq (1 - \gamma^2)^{-1/2} \|\Pi Q^\star - Q^\star\|$.*

The theorem establishes the convergence of LICRA to a stable point with the use of linear function approximators. The second statement bounds the proximity of the convergence point by the smallest approximation error that can be achieved given the choice of basis functions.

Having constructed a procedure to find the optimal agent's optimal value function, we now seek to determine the conditions when an intervention should be performed. Let us denote by $\{\tau_k\}_{k \geq 0}$ the points at which the agent decides to act or *intervention times*, so for example if the agent chooses to perform an action at state $s_6$ and again at state $s_8$, then $\tau_1 = 6$ and $\tau_2 = 8$. The following result characterises the optimal intervention policy $\mathfrak{g}$ and the optimal times $\{\tau_k\}_{k \geq 0}$.

**Proposition 1** *The policy $\mathfrak{g}$ is given by: $\mathfrak{g}(s_t) = H(\mathcal{M}^{\pi, \mathfrak{g}} Q^{\pi, \mathfrak{g}} - Q^{\pi, \mathfrak{g}})(s_t, a_t), \quad \forall s_t \in \mathcal{S}$, where $Q^{\pi, \mathfrak{g}}$ is the solution in Theorem 1, $\mathcal{M}$ is the intervention operator and $H$ is the Heaviside function, moreover the intervention times are $\tau_k = \inf\{\tau > \tau_{k-1} | \mathcal{M}^{\pi, \mathfrak{g}} Q^{\pi, \mathfrak{g}} = Q^{\pi, \mathfrak{g}}\}$.*

Prop. 1 characterises the (categorical) distribution $\mathfrak{g}$. Moreover, given the function $Q$, the times $\{\tau_k\}$ can be determined by evaluating if $\mathcal{M} Q = Q$ holds.

A key aspect of Prop. 1 is that it exploits the cost structure of the problem to determine when the agent should perform an intervention. In particular, the equality $\mathcal{M} Q = Q$ implies that performing an action and incurring a cost for doing so is optimal.

# 6 Budget Augmented LICRA via State Augmentation

We now tackle the problem of RL with a budget. To do this, we combine the above impulse control technology with state augmentation technique proposed in [28] The mathematical formulation of the problem is now given by the following for any $s \in \mathcal{S}$:

$$\max_{\pi \in \Pi, g} v^{\pi, \mathfrak{g}}(s) \text{ s. t. } n - \sum_{t=0}^{\infty} \sum_{k \geq 1} \delta_{\tau_k}^t \geq 0, \quad (5)$$

where $n \in \mathbb{N}$ is a fixed value that represents the maximum number of allowed interventions and $\sum_{k \geq 1} \delta_{\tau_k}^t$ is equal to one if an impulse was applied at time $t$ and zero if it was not. In order to avoid dealing with a constrained MDP, we propose to introduce a new variable $z_t$ tracking the remaining number of impulses: $z_t = n - \sum_{i=0}^{t-1} \sum_{k \geq 1} \delta_{\tau_k}^i$. We treat $z_t$ as another state and augment the state-space resulting in the transition $\widetilde{\mathcal{P}}$:

$$s_{t+1} \sim P(\cdot | s_t, a_t), \qquad z_{t+1} = z_t - \sum_{k \geq 1} \delta_{\tau_k}^t, \quad z_0 = n. \quad (6)$$

In order to avoid violations, we reshape the reward as follows: $\widetilde{\mathcal{R}}(s_t, z_t, a_t) = \begin{cases} \mathcal{R}(s_t, a_t) & z_t \geq 0, \\ -\Delta & z_t < 0, \end{cases}$

where $\Delta > 0$ is a large enough hyper-parameter ensuring that there are no safety violations. To summarise we aim to solve the following problem:

$$v^{\pi, \mathfrak{g}}(s, z) = \mathbb{E}\left[\sum_{t=0}^{\infty} \gamma^t \widetilde{\mathcal{R}}(s_t, z_t, a_t) | a_t \sim \pi(\cdot | s_t, z_t)\right], \tag{7}$$

where the policy now depends on the variable $z_t$. Note that $\widetilde{\mathcal{P}}$ in Equation 6 is a Markov process and, the rewards $\widetilde{\mathcal{R}}$ are bounded, as long as the rewards $\mathcal{R}$ are bounded. Therefore, we can apply directly the results for impulse control to this case as well. We denote the augmented MDP by $\widetilde{\mathcal{M}} = \langle \mathcal{S} \times \mathcal{Z}, \mathcal{A}, \widetilde{\mathcal{P}}, \widetilde{R}, \gamma \rangle$, where $\mathcal{Z}$ is the space of the augmented state. We have the following.

**Theorem 4** *Consider the MDP $\widetilde{\mathcal{M}}$ for the problem 7, then:*

*a) The Bellman equation holds, i.e. there exists a function $\tilde{v}^{*, \pi, \mathfrak{g}}$ s.th. $\tilde{v}^{*, \pi, \mathfrak{g}}(s, z) = \max_{a \in \mathcal{A}}\left(\widetilde{\mathcal{R}}(s, z, a) + \gamma \mathbb{E}_{s', z' \sim \mathcal{P}}\left[\tilde{v}^{*, \pi, \mathfrak{g}}(s', z')\right]\right)$, where the optimal policy for $\widetilde{\mathcal{M}}$ has the form $\pi^*(\cdot | s, z)$;*

*b) Given a $\tilde{v} : \mathcal{S} \times \mathcal{Z} \to \mathbb{R}$, the stable point solution for $\widetilde{\mathcal{M}}$ is a given by $\lim_{k \to \infty} \tilde{T}^k \tilde{v}^{\pi, g} = \max_{\hat{\pi} \in \Pi, \hat{\mathfrak{g}}} \tilde{v}^{\hat{\pi}, \hat{\mathfrak{g}}} = \tilde{v}^{*, \pi, \mathfrak{g}^*}$, where $(\pi^*, \mathfrak{g}^*)$ is an optimal policy of $\widetilde{\mathcal{M}}$ and $\tilde{T}$ is the Bellman operator of $\widetilde{\mathcal{M}}$.*

The result has several important implications. The first is that we can use a modified version of LICRA to obtain the solution of the problem while guaranteeing convergence (under standard assumptions). Secondly, our state augmentation procedure admits a Markovian representation of the optimal policy.

# 7 Experiments

We will now study empirically the performance of the LICRA framework. In experiments, we use different instances of LICRA, one where both policies are trained using PPO update (referred to as **LICRA_PPO**) and one where the policy deciding whether to act is trained using SAC and the other policy trained with PPO (referred to as **LICRA_SAC**). We have benchmarked both of these algorithms together with common baselines on environments, where it would be natural to introduce the concept of the cost associated with actions. We lastly performed a series of ablation studies which test LICRA's ability to handle different cost functions including the case when $c(s, a) \equiv 0$ which we defer to the Appendix which also contains further experiment details.

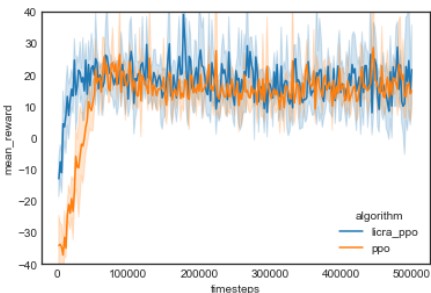

Figure 1: Training results in Merton investment problem for PPO style algorithms.

**Merton's Portfolio Problem with Transaction Costs.** Merton Investment Problem in which the investor faces transaction costs [10] is a well-known problem within finance. In our environment, the agent can decide to move its wealth between a risky asset and a risk-free asset. The agent receives a reward only at the final step, equal to the utility of the portfolio with a risk aversion factor equal to 0.5. If the final wealth of risky asset is $s_T$ and final wealth of risk-free asset is $c_T$, then the agent will receive a reward of $u(x) = 2\sqrt{s_T + c_T}$. The wealth evolves according to the following SDE:

$$dW_t = (r + p_t(\mu - r))W_t + W_t p_t \sigma dB_t \tag{8}$$

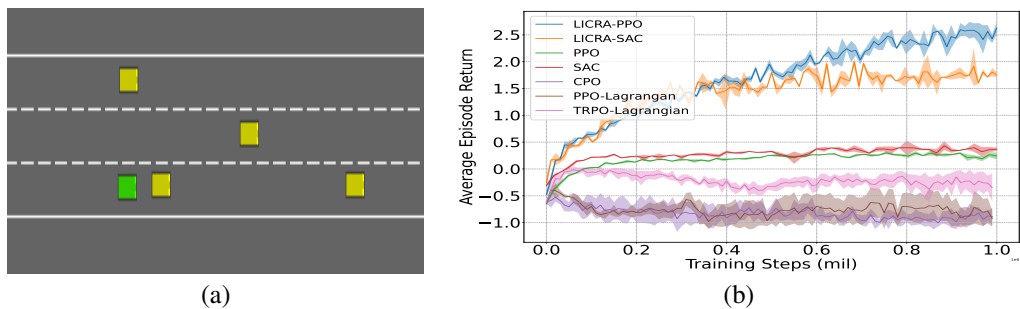

(a)                                             (b)

Figure 2: a) Drive Environment. b) Training results in drive environment.

where $W_t$ is the current wealth and the state variable, $dB_t$ is an increment of Brownian motion and $p_t$ is the proportion of wealth invested in the risky asset. We set the risk-free return $r = 0.01$, risky asset return $\mu = 0.05$ and volatility $\sigma = 1$. We discretise the action space so that at each step the agent has three actions available: move $10\%$ of risky asset wealth to the risk-free asset, move $10\%$ of risk-free asset wealth to the risky asset or do nothing. Each time the agent moves the assets, it incurs a cost of $1$ i.e. a transaction fee. The agent can act after a time interval of $0.01$ seconds and the episode ends after $75$ steps. The results of training are shown in Fig. 1 which clearly demonstrates that LICRA_PPO finds a better policy than standard PPO. Also comparing the variance among different seeds, we can see that LICRA_PPO is a much more stable algorithm than the other two.

**Driving Environment Fuel Rationing.** We studied an autonomous driving scenario where fuel-efficient driving is a priority. One of the main components of fuel-efficient driving is controlled usage of acceleration and braking, in the sense that 1) the amount of acceleration and braking should be limited 2) if accelerations should be performed slowly and gently. We believe this is a problem where LICRA should thrive as the impulse control agent can learn to restrict the amount of acceleration and braking in the presence of other cars and choose when to allow the car to decelerate naturally. We used the highway-env [17] environment on a highway task (see Fig (2. a)) where the green vehicle is our controlled vehicle and the goal is to avoid crashing into other vehicles whilst driving at a reasonable speed. We add a cost function into the reward term dependent on the continuous acceleration action, $C(a_t) = K + a_t^2$, where $K > 0$ is a fixed constant cost of taking any action, and $a_t \in [-1, 1]$, with larger values of acceleration or braking being penalised more. The results are presented in Fig. (2.b). Notably, LICRA is able to massively outperform the baselines, especially our safety specific baselines which struggle to deal with the cost function associated with the environment. We believe one reason for the success of LICRA is that it is far easier for it to utilise the null action of zero acceleration/braking than the other algorithms, whilst all the algorithms have a guaranteed cost at every time step whilst not gaining a sizeable reward to counter the cost.

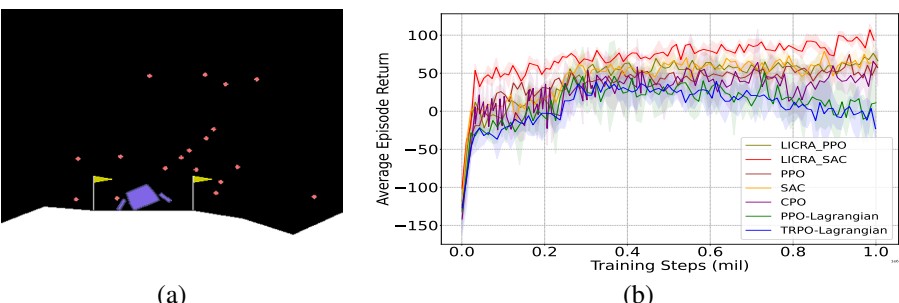

(a)                                             (b)

Figure 3: a) The lander must land on the pad between two flags. . b) Training results in Lunar Lander.

**Lunar Lander Environment.** We tested the ability of LICRA to perform in environment that simulate real-world physical dynamics. We tested LICRA's performance the Lunar Lander environment in OpenAI gym [7] which we adjusted to incorporate minimal bounded costs in the reward definition. In this environment, the agent is required to maintain both a good posture mid-air and reach the landing pad as quickly as possible. The reward function is given by:

$$\text{Reward}(s_t) = 3 * (1 - \mathbf{1}_{d_t - d_{t-1} = 0}) - 3 * (1 - \mathbf{1}_{v_t - v_{t-1} = 0}) - 3 * (1 - \mathbf{1}_{\omega_t - \omega_{t-1} = 0})$$
$$-0.03 * \text{FuelSpent}(s_t) - 10 * (v_t - v_{t-1}) - 10 * (\omega_t - \omega_{t-1}) + 100 * \text{hasLanded}(s_t)$$

where $d_t$ is the distance to the landing pad, $v_t$ is the velocity of the agent, and $\omega_t$ is the angular velocity of the agent at time $t$. $\mathbf{1}_X$ is the indicator function of taking actions, which is 1 when the statement $X$ is true and 0 when $X$ is false. Considering the limited fuel budget, we assume that we have a fixed cost for each action taken by the agent here, and doing nothing brings no cost. Then, to describe the goal of the game, we define the function of the final status by hasLanded(), which is 0 when not landing; 1 when the agent has landed softly on the landing pad; and $-1$ when the lander runs out of fuel or loses contact with the pad on landing. The reward function rewards the agent for reducing its distance to the landing pad, decreasing its speed to land smoothly and keeping the angular speed at a minimum to prevent rolling. Additionally, it penalises the agent for running out of fuel and deters the agent from taking off again after landing.

By introducing a reward function with minimally bounded costs, our goal was to test if LICRA can exploit the optimal policy. In Fig. 3, we observe that the LICRA agent outperforms all the baselines, both in terms of sample efficiency and average test return (total rewards at each timestep). We also observe that LICRA enables more stable training than PPO, PPO-Lagrangian and CPO.

**Ablation Study 1. Prioritisation of Most Important Actions.** We next tested LICRA's ability to prioritise where it performs actions when the necessity to act varies significantly between states. To test this, we modified the Drive Environment to now consist of a single lane, a start state and a goal state start (at the end) where there is a reward. With no acceleration, the vehicle decreases velocity. To reach the goal, the agent must apply an acceleration $a_t \in [-1, 1]$. Each acceleration $a_t$ incurs a cost $C(a_t)$ as defined above. At zones $k = 1, 2, 3$ of the lane, if the vehicle is travelling below a velocity $v_{min}$, it is penalised by a strictly negative cost $c_k$ where $c_1 < c_2 < c_3$. As shown in Fig. 4, when the intervention cost increases i.e. when $K \to \infty$, LICRA successfully prioritises the highest penalty zones to avoid incurring large costs.

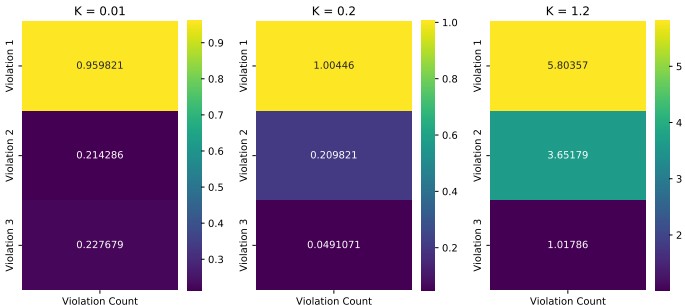

Figure 4: Results for Ablation Study 1. Heatmaps display the number of times the agent drives below $v_{min}$ in the penalty zones. Violation 1 refers to the lowest cost zone, whilst Violation 3 refers to the largest cost zone. $K$ refers to the fixed cost for taking an action.

# 8 Conclusion

We presented a novel method to tackle the problem of learning how to select when to act in addition to learning which actions to execute. Our framework, which is a general tool for tackling problems of this kind seamlessly adopts RL algorithms enabling them to efficiently tackle problems in which the agent must be selective about when it executes actions. This is of fundamental importance in practical settings where performing many actions over the horizon can lead to costs and undermine the service life of machinery. We demonstrated that our solution, LICRA which at its core has a sequential decision structure that first decides whether or not an action ought to be taken under the action policy can solve tasks where the agent faces costs with extreme efficiency as compared to leading reinforcement learning methods. In some tasks, we showed that LICRA is able to solve problems that are unsolvable using current reinforcement learning machinery. We envisge that this framework can serve as the basis extensions to different settings including adversarial training for solving a variety of problems within RL.

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
