# Part I

# Appendix

## Table of Contents

# 9 Full Algorithms

In this supplementary section, we explicitly define two versions of the LICRA algorithm. Algorithm 2 describes the version where both agents use PPO. PPO_update() subroutine is a standard PPO gradient update done as in Algorithm 1 of [27] with clipping surrogate objective with parameter $\epsilon$. The gradient update utilises batch size $B$, stepsize $\alpha$ and performs $T$ update steps per episode. Algorithm 3 defines the LICRA_SAC version, utilising SAC for the switching agent. Here SAC_update() is analogously a standard soft actor-critic update done as in Algorithm 1 of [14], where $B$, $\alpha$ and $T$ play the same role as in the PPO update.

---

**Algorithm 2:** LICRA with PPO

---

1: **Input:** Stepsize $\alpha$, batch size $B$, episodes $K$, steps per episode $T$, mini-epochs $e$, clipping-parameter $\epsilon$
2: **Initialise:** Policy network (acting) $\pi$, Policy network (switching) $\mathfrak{g}$, Critic network (acting)$V_\pi$,Critic network (switching)$V_\mathfrak{g}$
3: Given reward objective function, $R$, initialise Rollout Buffers $\mathcal{B}_\pi, \mathcal{B}_\mathfrak{g}$
4: **for** $N_{episodes}$ **do**
5:     Reset state $s_0$, Reset Rollout Buffers $\mathcal{B}_\pi, \mathcal{B}_\mathfrak{g}$
6:     **for** $t = 0, 1, \ldots$ **do**
7:         Sample $a_t \sim \pi(\cdot|s_t)$
8:         Sample $g_t \sim \mathfrak{g}(\cdot|s_t)$
9:         **if** $g_t = 0$ **then**
10:            Apply $a_t$ so $s_{t+1} \sim P(\cdot|a_t, s_t)$,
11:            Receive rewards $r_t = \mathcal{R}(s_t, a_t)$
12:            Store $(s_t, a_t, s_{t+1}, r_t)$ in $\mathcal{B}_\pi$
13:         **else**
14:            Apply the null action so $s_{t+1} \sim P(\cdot|0, s_t)$,
15:            Receive rewards $r_t = \mathcal{R}(s_t, 0)$.
16:         **end if**
17:         Store $(s_t, g_t, s_{t+1}, r_t)$ in $\mathcal{B}_\mathfrak{g}$
18:     **end for**
19:     **// Learn the individual policies**
20:     PPO_update($\pi, V_\pi, \mathfrak{B}_\pi, B, \alpha, T$)
21:     PPO_update($\mathfrak{g}, V_\mathfrak{g}, \mathfrak{B}_\mathfrak{g}, B, \alpha, T$)
22: **end for**

---

**Algorithm 3:** LICRA with SAC

1: **Input:** Stepsize $\alpha$, batch size $B$, episodes $K$, steps per episode $T$, mini-epochs $e$
2: **Initialise:** Policy network (acting) $\pi$, Policy network (switching) $\mathfrak{g}$,
   Critic network (acting)$V_\pi$,Q-Critic network (switching )$Q_{\mathfrak{g}}$,V-Critic network (switching)$V_{\mathfrak{g}}$
3: Given reward objective function, $R$, initialise Rollout Buffer $\mathcal{B}_\pi$ Replay Buffer $\mathcal{B}_{\mathfrak{g}}$
4: **for** $N_{episodes}$ **do**
5:    Reset state $s_0$, Reset Rollout Buffer $\mathcal{B}_\pi$
6:    **for** $t = 0, 1, \ldots$ **do**
7:       Sample $a_t \sim \pi(\cdot|s_t)$
8:       Sample $g_t \sim \mathfrak{g}(\cdot|s_t)$
9:       **if** $g_t = 0$ **then**
10:          Apply $a_t$ so $s_{t+1} \sim P(\cdot|a_t, s_t)$,
11:          Receive rewards $r_t = \mathcal{R}(s_t, a_t)$
12:          Store $(s_t, a_t, s_{t+1}, r_t)$ in $\mathcal{B}_\pi$
13:       **else**
14:          Apply the null action so $s_{t+1} \sim P(\cdot|0, s_t)$,
15:          Receive rewards $r_t = \mathcal{R}(s_t, 0)$.
16:       **end if**
17:       Store $(s_t, g_t, s_{t+1}, r_t)$ in $\mathcal{B}_{\mathfrak{g}}$
18:    **end for**
19:    **// Learn the individual policies**
20:    PPO_update($\pi$, $V_\pi$, $\mathfrak{B}_\pi$, $B$, $\alpha$, $T$)
21:    Sample a batch of $|B_{\mathfrak{g}}|$ transitions $B_{\mathfrak{g}}$ from $\mathcal{B}_{\mathfrak{g}}$
22:    SAC_update($\mathfrak{g}$, $V_{\mathfrak{g}}$, $Q_{\mathfrak{g}}$ , $B_{\mathfrak{g}}$, $B$, $\alpha$, $T$)
23: **end for**

## 10  Fuel Budget

We test the ability of LICRA (LICRA_SAC) in a budgeted Drive environment. We modified the Drive environment in Sec. 7 so that there is an additional scarce fuel level for the controlled vehicle. This fuel level decreases in proportion to the magnitude of the action taken by the agent. If the controlled vehicle runs out fuel then it receives a large negative reward, and therefore it is important that the agent learns to use minimal acceleration/braking to reach the final destination.

Fig. 5 shows the performance of LICRA and corresponding baselines. The majority of the baselines fail to learn anything in the environment, which we expect is due to the difficult of exploration due to low fuel levels (LICRA can counter this by exploring using the null action) and how easy it is to run out of fuel. In some seeds CPO is able to solve the environment to the same level as LICRA, but is not as stable over seeds and it is not as fast as LICRA in improving performance.

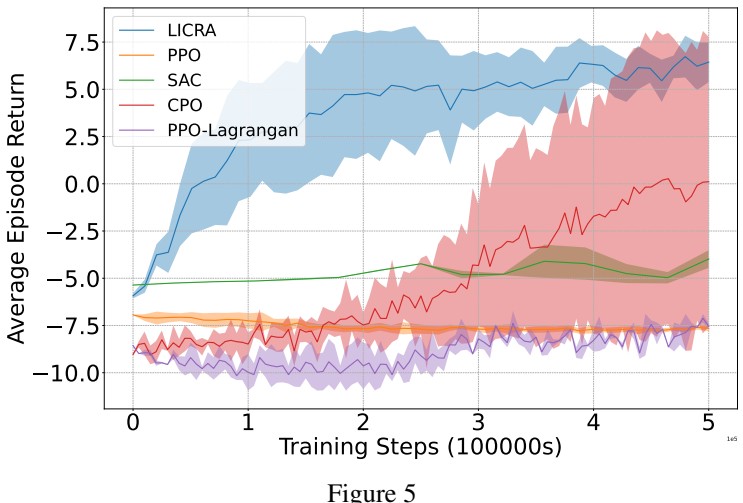

Figure 5

## 11  Ablation studies

Fig. 3 shows the performance of LICRA variants and baselines when $c(s, a) \equiv 5$. In this section, we analyse LICRA's ability to handle various functional forms of the intervention cost.

### 11.1  Ablation Study 2: Testing Different functional forms of the intervention costs

***Intervention costs of the form $c(s, a) \equiv 0$.***

In Sec. 4, we claimed that LICRA's impulse control mechanism which first determines the set of states to perform actions then only learns the optimal actions at these states can induce a much quicker learning proceess. In particular, we claimed that LICRA enables the RL agent to rapidly learns which states to focus on to learn optimal actions.

In our last ablation analysis, we test LICRA's ability to handle the case in which the agent faces no fixed costs so $c(s, a) \equiv 0$, therefore deviating from the form of the impulse control objective (1). In doing so, we test the ability of LICRA to prioritise the most important states for learning the correct actions in a general RL setting. As shown in Fig. 6 we present the average returns when $c(s, a) \equiv 0$. For this case, LICRA_SAC both learns the fastest indicating the benefits of the impulse control framework even in the setting in which the agent does not face fixed minimal costs for each action. Strikingly, LICRA also achieves the highest performance which demonstrates that LICRA also improves overall performance in complex tasks.

***Intervention costs of the form $c(s, a) \equiv k > 0$.***

We first analyse the behaviour of LICRA and leading baselines when the environment has intervention costs of the form $c(s, a) = k$ where the fixed part $k > 0$ is a strictly positive constant. Intervention

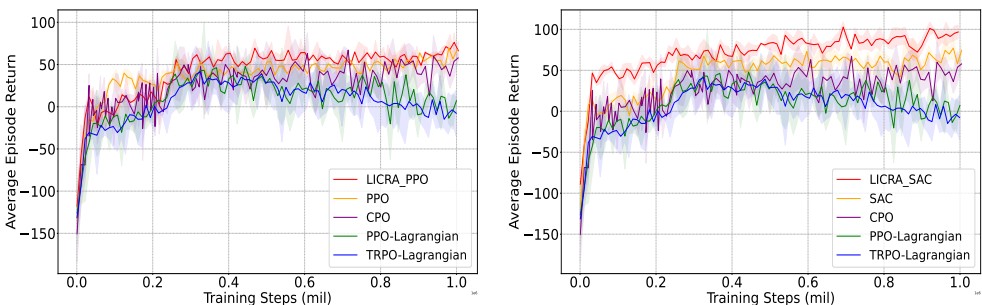

Figure 6: Ablation Study when $c(s, a) \equiv 0$ in Lunar Lander.

costs of this kind are frequently found within economic settings where they are characterised as 'menu costs'. This name derives from the fixed costs associated to a vendor adjusting their prices and serves as an explanation for price rigidities within the macroeconomy [8, 21].

We next test the average returns when $c(s, a) \equiv 10$, $c(s, a) \equiv 20$, as shown in Fig. 7. Note that the action space is discrete and the intervention costs only occur when $a \neq 0$. In this case, LICRA_SAC both learns the fastest and achieves the highest performance. Moreover, unlike PPO which produces declining performance and does not converge, LICRA_PPO converges to a high reward stable point.

***Intervention costs of the form $c(s, a) = k + \lambda a$.***

We next analyse the behaviour of LICRA and leading baselines when the environment has intervention costs of the form $c(s, a) = k + \lambda a$ where the fixed part and proportional part $\lambda, k > 0$ are strictly positive constants. Intervention costs of this kind are frequently found within financial settings in which an investor incurs a fixed cost for investment decisions e.g. broker costs [10, 20]. Note that the action space is discrete and the intervention costs only occur when $a \neq 0$.

We present the average returns when $c(s, a) = 5 + |a|$, $c(s, a) = 5 + 5 \cdot |a|$ in Fig. 7. As with previous case, LICRA_SAC both learns the fastest and achieves the highest performance. Moreover, unlike PPO which produces declining performance and does not converge, LICRA_PPO converges to a high reward stable point.

***Intervention costs of the form $c(s, a) = k + f(s, a)$.***

In general, the intervention costs incurred by an agent for each intervention can be allowed to be a function of the state. For example, activating an actuator under adverse environment conditions may incur greater wear to machinery than in other conditions. The functional form of the cost function is the most general and produces the most complex decision problem out of the aforementioned cases. To capture this general case, we lastly analysed the behaviour of LICRA and leading baselines when the environment has intervention costs of the form $c(s, a) = k + f(s, a)$ where the fixed part $k > 0$ and $f : \mathcal{S} \times \mathcal{A} \to \mathbb{R}_{>0}$ is a positive function.

As shown in Fig. 7, for this case we present the average returns when $c(s, a) = 5 + |d_s - d_{\text{target}}| \cdot |a|$, where $|d_s - d_{\text{target}}|$ represents the distance between the current position to the destination (determined by the state $s$). As before, the action space is discrete and the intervention costs only occur when $a \neq 0$. As with previous cases, LICRA_SAC both learns the fastest and achieves the highest performance, demonstrating LICRA's ability to solve the more complex task. Moreover, as in the previous cases, unlike PPO which produces declining performance and does not converge, LICRA_PPO converges to a high reward stable point.

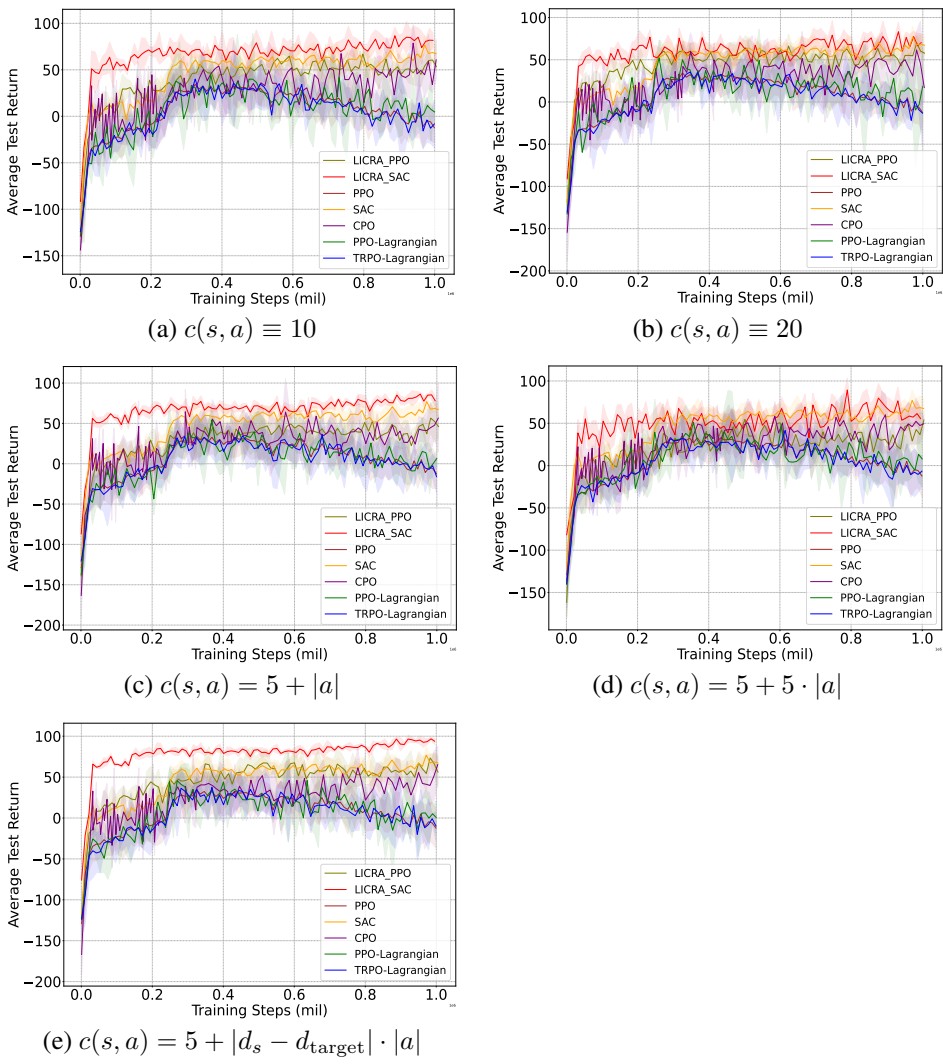

(a) $c(s, a) \equiv 10$

(b) $c(s, a) \equiv 20$

(c) $c(s, a) = 5 + |a|$

(d) $c(s, a) = 5 + 5 \cdot |a|$

(e) $c(s, a) = 5 + |d_s - d_{\text{target}}| \cdot |a|$

Figure 7: Ablation Study when (a) $c(s, a) \equiv 10$, (b) $c(s, a) \equiv 20$, (c) $c(s, a) = 5 + |a|$, (d) $c(s, a) = 5 + 5 \cdot |a|$ and (e) $c(s, a) = 5 + |d_s - d_{\text{target}}| \cdot |a|$ in Lunar Lander.

## 11.2 Ablation Study 3: Benefits of LICRA in Smaller Intervention Region

In Sec. 4, we claimed that LICRA enables efficient learning in setting in which the set of states in which the agent should act, which we call the *intervention region* is a subset of the state space. Moreover, we claimed that this advantage over existing methods is increased as the intervention region becomes small in relation to the entire state space.

To test these claims, we modified the Drive environment in Sec. 7 to a problem setting in which the agent is required to bring a moving vehicle to rest in a particular subregion of the lane or *stop gap* which we denote by $\mathcal{S}_I \subset \mathcal{S}$. If the agent brings the vehicle to rest within $\mathcal{S}_I$ the agent receives a reward $R > 0$ and the episode terminates. If however the agent brings the vehicle to rest outside of $\mathcal{S}_I$ the agent receives a lower reward $r < R$ the episode terminates. Lastly, if the agent fails to bring the vehicle to rest before the end of the lane the episode terminates and the agent receives 0 reward. The length of the entire lane $\mathcal{S}$ is 500 units and we ablate the size of the region $\mathcal{S}_I$ in which the agent is required to stop to receive the maximum reward. Now the agent gets to decide a magnitude which decelerates the vehicle i.e. how heavily to brake (at any given point, the agent can also choose not to brake at all). Each deceleration $a \in [0, 1]$ incurs a fixed minimal cost of $c(s, a) = \kappa + \lambda a$ where $\kappa, \lambda > 0$.

Fig. 8 shows the results of the ablation on the stop gap $\mathcal{S}_I$ when $\mathcal{S}_I$ is a length of 50 units or 10% of the entire state space $\mathcal{S}$ through to $\mathcal{S}_I \equiv \mathcal{S}$ i.e. when the intervention region is the entire state space.

As can be observed in Fig. 8, when the intervention region is comparatively small, LICRA_PPO produces a significant performance advantage over the base learner PPO. This performance gap is gradually decreased as the size of the stop gap increases and eventually becomes the entire state space (which is the case when the stop gap is 500). Interestingly, LICRA_PPO still maintains a performance advantage over PPO even when $\mathcal{S}_I \equiv \mathcal{S}$.

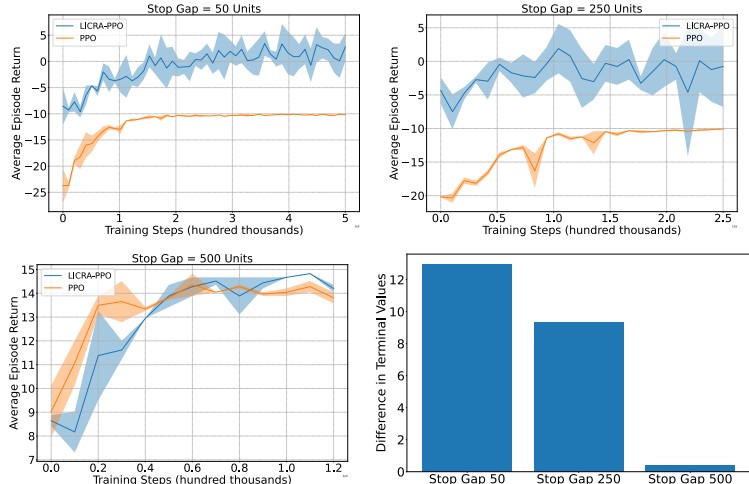

Figure 8: Ablation on the intervention region. The 'Stop Gap' represents the intervention region in our modified Drive environment. When the Stop Gap is 50 units, the intervention region is 10% of the entire state space. When the Stop Gap is 509 units, the intervention region is the entire state space.

## 12    Analysis of LICRA Q-learning Variant

In order to validate the convergence of the Q learning variant of LICRA (c.f. (4)), we ran an experiment where the LICRA Q learning variant given in (4) can choose whether to act or not to act in a given discrete state (and continuous action space). In keeping with the problem setting we consider, there is a cost associated with any non-zero action. LICRA decides whether to act or not to act based on a tabular Q-learning rule given in (4) (using a normalised advantage function (NAF) to handle the continuous action space), where we store expected value for non acting in each cell. Fig. 9 shows that the TD-error of this tabular Q learning setting converges to zero which validates the results of Theorem 2 and Theorem 3.

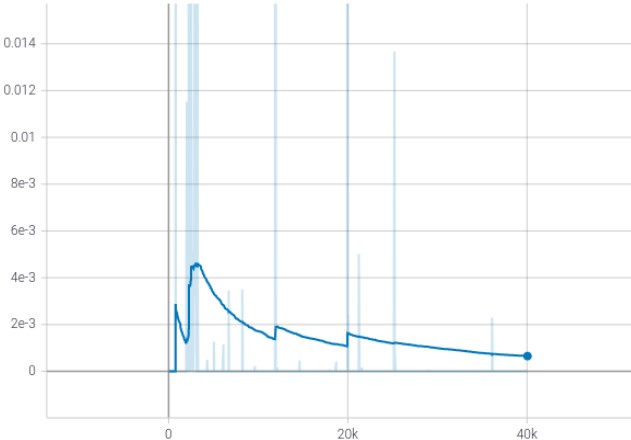

Figure 9: TD error of tabular version of LICRA (smoothed).

## 13 Hyperparameter Settings

In the table below we report all hyperparameters used in Merton Investment problem experiments.

| Hyperparameter | Value (PPO methods) | Value (SAC) |
|---|---|---|
| Clip Gradient Norm | 0.5 | None |
| Discount Factor | 0.99 | 0.99 |
| Learning rate | $1\text{x}10^{-3}$ | $1\text{x}10^{-4}$ |
| Batch size | 32 | 1024 |
| Steps per epoch | 2000 | 2000 |
| Optimisation algorithm | ADAM | ADAM |

In the next table, we report hyperparameters used in remaining experiments.

| Hyperparameter | Value |
|---|---|
| Clip Gradient Norm | 1 |
| $\gamma_E$ | 0.99 |
| $\lambda$ | 0.95 |
| Learning rate | $1\text{x}10^{-4}$ |
| Number of minibatches | 4 |
| Number of optimisation epochs | 4 |
| Number of parallel actors | 16 |
| Optimisation algorithm | ADAM |
| Rollout length | 128 |
| Sticky action probability | 0.25 |
| Use Generalized Advantage Estimation | True |

## 14  Notation & Assumptions

We assume that $\mathcal{S}$ is defined on a probability space $(\Omega, \mathcal{F}, \mathbb{P})$ and any $s \in \mathcal{S}$ is measurable with respect to the Borel $\sigma$-algebra associated with $\mathbb{R}^p$. We denote the $\sigma$-algebra of events generated by $\{s_t\}_{t \geq 0}$ by $\mathcal{F}_t \subset \mathcal{F}$. In what follows, we denote by $(\mathcal{V}, \|\|)$ any finite normed vector space and by $\mathcal{H}$ the set of all measurable functions.

The results of the paper are built under the following assumptions which are standard within RL and stochastic approximation methods:

**Assumption 1.** The stochastic process governing the system dynamics is ergodic, that is the process is stationary and every invariant random variable of $\{s_t\}_{t \geq 0}$ is equal to a constant with probability $1$.

**Assumption 2.** The function $R$ is in $L_2$.

**Assumption 3.** For any positive scalar $c$, there exists a scalar $\mu_c$ such that for all $s \in \mathcal{S}$ and for any $t \in \mathbb{N}$ we have: $\mathbb{E}\left[1 + \|s_t\|^c | s_0 = s\right] \leq \mu_c(1 + \|s\|^c)$.

**Assumption 4.** There exists scalars $C_1$ and $c_1$ such that for any function $J$ satisfying $|v(s)| \leq C_2(1 + \|s\|^{c_2})$ for some scalars $c_2$ and $C_2$ we have that: $\sum_{t=0}^{\infty} |\mathbb{E}\left[v(s_t)|s_0 = s\right] - \mathbb{E}[v(s_0)]| \leq C_1 C_2(1 + \|s_t\|^{c_1 c_2})$.

**Assumption 5.** There exists scalars $c$ and $C$ such that for any $s \in \mathcal{S}$ we have that: $|\mathcal{R}(s, \cdot)| \leq C(1 + \|s\|^c)$.

## 15  Proof of Technical Results

We begin the analysis with some preliminary lemmata and definitions which are useful for proving the main results.

**Definition 1** *A.1 An operator $T : \mathcal{V} \to \mathcal{V}$ is said to be a **contraction** w.r.t a norm $\| \cdot \|$ if there exists a constant $c \in [0, 1[$ such that for any $V_1, V_2 \in \mathcal{V}$ we have that:*

$$\|TV_1 - TV_2\| \leq c\|V_1 - V_2\|. \tag{9}$$

**Definition 2** *A.2 An operator $T : \mathcal{V} \to \mathcal{V}$ is **non-expansive** if $\forall V_1, V_2 \in \mathcal{V}$ we have:*

$$\|TV_1 - TV_2\| \leq \|V_1 - V_2\|. \tag{10}$$

**Lemma 2** *For any $f : \mathcal{V} \to \mathbb{R}, g : \mathcal{V} \to \mathbb{R}$, we have that:*

$$\left\| \max_{a \in \mathcal{V}} f(a) - \max_{a \in \mathcal{V}} g(a) \right\| \leq \max_{a \in \mathcal{V}} \|f(a) - g(a)\| . \tag{11}$$

**Proof 1** *We restate the proof given in [23]:*

$$f(a) \leq \|f(a) - g(a)\| + g(a) \tag{12}$$

$$\implies \max_{a \in \mathcal{V}} f(a) \leq \max_{a \in \mathcal{V}}\{\|f(a) - g(a)\| + g(a)\} \leq \max_{a \in \mathcal{V}} \|f(a) - g(a)\| + \max_{a \in \mathcal{V}} g(a). \tag{13}$$

*Deducting $\max_{a \in \mathcal{V}} g(a)$ from both sides of (13) yields:*

$$\max_{a \in \mathcal{V}} f(a) - \max_{a \in \mathcal{V}} g(a) \leq \max_{a \in \mathcal{V}} \|f(a) - g(a)\| . \tag{14}$$

*After reversing the roles of $f$ and $g$ and redoing steps (12) - (13), we deduce the desired result since the RHS of (14) is unchanged.*

**Lemma 3** *A.4 The probability transition kernel $P$ is non-expansive, that is:*

$$\|PV_1 - PV_2\| \leq \|V_1 - V_2\|. \tag{15}$$

**Proof 2** *The result is well-known e.g. [31]. We give a proof using the Tonelli-Fubini theorem and the iterated law of expectations, we have that:*

$$\|PJ\|^2 = \mathbb{E}\left[(PJ)^2[s_0]\right] = \mathbb{E}\left(\left[\mathbb{E}\left[J[s_1]|s_0\right]\right)^2\right) \leq \mathbb{E}\left[\mathbb{E}\left[J^2[s_1]|s_0\right]\right] = \mathbb{E}\left[J^2[s_1]\right] = \|J\|^2,$$

*where we have used Jensen's inequality to generate the inequality. This completes the proof.*

 **Proof of Theorem 1**

574 **Proof 3 (Proof of Lemma 1)** *Recall we define the Bellman operator $T$ acting on a function $v^{\pi,\mathfrak{g}}$ :*
575 $\mathcal{S} \to \mathbb{R}$ *by*

$$Tv^{\pi,\mathfrak{g}}(s_{\tau_k}) := \max \left\{ \mathcal{M}^{\pi,\mathfrak{g}} Q^{\pi,\mathfrak{g}}(s_{\tau_k}, a_{\tau_k}), \left[ \mathcal{R}(s_{\tau_k}, 0) + \gamma \sum_{s' \in \mathcal{S}} P(s'; 0, s_{\tau_k}) v^{\pi,\mathfrak{g}}(s') \right] \right\} \quad (16)$$

576 *In what follows and for the remainder of the script, we employ the following shorthands:*

$$\mathcal{P}_{ss'}^a =: \sum_{s' \in \mathcal{S}} P(s'; a, s), \quad \mathcal{P}_{ss'}^\pi =: \sum_{a \in \mathcal{A}} \pi(a|s) \mathcal{P}_{ss'}^a, \quad \mathcal{R}^\pi(s_t) := \sum_{a_t \in \mathcal{A}} \pi(a_t|s) R(s_t, a_t)$$

577 *For notational simplicity, where it will not cause confusion we also drop the dependence of the*
578 *functions $v^{\pi,\mathfrak{g}}, Q^{\pi,\mathfrak{g}}$ on the policy pair $(\pi, \mathfrak{g})$ and with a slight abuse of notation we will write*
579 $\mathcal{M}v^{\pi',\mathfrak{g}'}(s_{\tau_k}) := \max_{a \in \mathcal{A}} \left\{ \mathcal{R}(s_{\tau_k}, a) - c(s_{\tau_k}, a) + \gamma \sum_{s' \in \mathcal{S}} P(s'; a, s) v^{\pi',\mathfrak{g}'}(s') \right\}.$

580 *To prove that $T$ is a contraction, we consider the three cases produced by (16), that is to say we prove*
581 *the following statements:*

582 *i)* $\left| \mathcal{R}(s_t, a_t) + \gamma \max_{a \in \mathcal{A}} \mathcal{P}_{s's_t}^a v(s') - \left( \mathcal{R}(s_t, a_t) + \gamma \max_{a \in \mathcal{A}} \mathcal{P}_{s's_t}^a v'(s') \right) \right| \leq \gamma \|v - v'\|$

583 *ii)* $\|\mathcal{M}v - \mathcal{M}v'\| \leq \gamma \|v - v'\|,$      *(and hence $\mathcal{M}$ is a contraction).*

584 *iii)* $\left\| \mathcal{M}v - \left[ \mathcal{R}(\cdot, a) + \gamma \max_{a \in \mathcal{A}} \mathcal{P}^a v' \right] \right\| \leq \gamma \|v - v'\|.$

585 *We begin by proving i).*

586 *Indeed, for any $a \in \mathcal{A}$ and $\forall s_t \in \mathcal{S}, \forall s' \in \mathcal{S}$ we have that*

$$\left| \mathcal{R}(s_t, a_t) + \gamma \max_{a \in \mathcal{A}} \mathcal{P}_{s's_t}^a v(s') - \left[ \mathcal{R}(s_t, a_t) + \gamma \max_{a \in \mathcal{A}} \mathcal{P}_{s's_t}^a v'(s') \right] \right|$$
$$\leq \max_{a \in \mathcal{A}} \left| \gamma \mathcal{P}_{s's_t}^a v(s') - \gamma \mathcal{P}_{s's_t}^a v'(s') \right|$$
$$\leq \gamma \|Pv - Pv'\|$$
$$\leq \gamma \|v - v'\|,$$

587 *again using the fact that $P$ is non-expansive and Lemma 2.*

588 *We now prove ii).*

589 *For any $\tau \in \mathcal{F}$, define by $\tau' = \inf\{t > \tau | s_t \in \mathcal{S}_I, \tau \in \mathcal{F}_t\}$. Now using the definition of $\mathcal{M}$ we have*
590 *that for any $s_\tau \in \mathcal{S}$*

$$|(\mathcal{M}v - \mathcal{M}v')(s_\tau)|$$
$$\leq \max_{a_\tau \in \mathcal{A}} \left| \mathcal{R}(s_\tau, a_\tau) + c(s_\tau, a_\tau) + \gamma \mathcal{P}_{s's_\tau}^\pi \mathcal{P}^a v(s_\tau) - \left( \mathcal{R}(s_\tau, a_\tau) + c(s_\tau, a_\tau) + \gamma \mathcal{P}_{s's_\tau}^\pi \mathcal{P}^a v'(s_\tau) \right) \right|$$
$$= \gamma \left| \mathcal{P}_{s's_\tau}^\pi \mathcal{P}^a v(s_\tau) - \mathcal{P}_{s's_\tau}^\pi \mathcal{P}^a v'(s_\tau) \right|$$
$$\leq \gamma \|Pv - Pv'\|$$
$$\leq \gamma \|v - v'\|,$$

591 *using the fact that $P$ is non-expansive. The result can then be deduced easily by applying max on*
592 *both sides.*

593 *We now prove iii). We split the proof of the statement into two cases:*

594 *Case 1:*

$$\mathcal{M}v(s_\tau) - \left( \mathcal{R}(s_\tau, a_\tau) + \gamma \max_{a \in \mathcal{A}} \mathcal{P}_{s's_\tau}^a v'(s') \right) < 0. \quad (17)$$

*We now observe the following:*

$$\mathcal{M}v(s_\tau) - \mathcal{R}(s_\tau, a_\tau) + \gamma \max_{a \in \mathcal{A}} \mathcal{P}^a_{s's_\tau} v'(s')$$

$$\leq \max\left\{\mathcal{R}(s_\tau, a_\tau) + \gamma \mathcal{P}^\pi_{s's_\tau} \mathcal{P}^a v(s'), \mathcal{M}v(s_\tau)\right\} - \mathcal{R}(s_\tau, a_\tau) + \gamma \max_{a \in \mathcal{A}} \mathcal{P}^a_{s's_\tau} v'(s')$$

$$\leq \left| \max\left\{\mathcal{R}(s_\tau, a_\tau) + \gamma \mathcal{P}^\pi_{s's_\tau} \mathcal{P}^a v(s'), \mathcal{M}v(s_\tau)\right\} - \max\left\{\mathcal{R}(s_\tau, a_\tau) + \gamma \max_{a \in \mathcal{A}} \mathcal{P}^a_{s's_\tau} v'(s'), \mathcal{M}v(s_\tau)\right\} \right.$$

$$\left. + \max\left\{\mathcal{R}(s_\tau, a_\tau) + \gamma \max_{a \in \mathcal{A}} \mathcal{P}^a_{s's_\tau} v'(s'), \mathcal{M}v(s_\tau)\right\} - \mathcal{R}(s_\tau, a_\tau) + \gamma \max_{a \in \mathcal{A}} \mathcal{P}^a_{s's_\tau} v'(s') \right|$$

$$\leq \left| \max\left\{\mathcal{R}(s_\tau, a_\tau) + \gamma \max_{a \in \mathcal{A}} \mathcal{P}^a_{s's_\tau} v(s'), \mathcal{M}v(s_\tau)\right\} - \max\left\{\mathcal{R}(s_\tau, a_\tau) + \gamma \max_{a \in \mathcal{A}} \mathcal{P}^a_{s's_\tau} v'(s'), \mathcal{M}v(s_\tau)\right\} \right|$$

$$+ \left| \max\left\{\mathcal{R}(s_\tau, a_\tau) + \gamma \max_{a \in \mathcal{A}} \mathcal{P}^a_{s's_\tau} v'(s'), \mathcal{M}v(s_\tau)\right\} - \mathcal{R}(s_\tau, a_\tau) + \gamma \max_{a \in \mathcal{A}} \mathcal{P}^a_{s's_\tau} v'(s') \right|$$

$$\leq \gamma \max_{a \in \mathcal{A}} \left| \mathcal{P}^\pi_{s's_\tau} \mathcal{P}^a v(s') - \mathcal{P}^\pi_{s's_\tau} \mathcal{P}^a v'(s') \right|$$

$$+ \left| \max\left\{0, \mathcal{M}v(s_\tau) - \left(\mathcal{R}(s_\tau, a_\tau) + \gamma \max_{a \in \mathcal{A}} \mathcal{P}^a_{s's_\tau} v'(s')\right)\right\} \right|$$

$$\leq \gamma \|Pv - Pv'\|$$

$$\leq \gamma \|v - v'\|,$$

*where we have used the fact that for any scalars $a, b, c$ we have that $|\max\{a, b\} - \max\{b, c\}| \leq |a - c|$ and the non-expansiveness of $P$.*

**Case 2:**

$$\mathcal{M}v(s_\tau) - \left(\mathcal{R}(s_\tau, a_\tau) + \gamma \max_{a \in \mathcal{A}} \mathcal{P}^a_{s's_\tau} v'(s')\right) \geq 0.$$

*For this case, first recall that for any $\tau \in \mathcal{F}$, $-c(s_\tau, a_\tau) > \lambda$ for some $\lambda > 0$.*

$$\mathcal{M}v(s_\tau) - \left(\mathcal{R}(s_\tau, a_\tau) + \gamma \max_{a \in \mathcal{A}} \mathcal{P}^a_{s's_\tau} v'(s')\right)$$

$$\leq \mathcal{M}v(s_\tau) - \left(\mathcal{R}(s_\tau, a_\tau) + \gamma \max_{a \in \mathcal{A}} \mathcal{P}^a_{s's_\tau} v'(s')\right) - c(s_\tau, a_\tau)$$

$$\leq \mathcal{R}(s_\tau, a_\tau) + c(s_\tau, a_\tau) + \gamma \mathcal{P}^\pi_{s's_\tau} \mathcal{P}^a v(s')$$

$$- \left(\mathcal{R}(s_\tau, a_\tau) + c(s_\tau, a_\tau) + \gamma \max_{a \in \mathcal{A}} \mathcal{P}^a_{s's_\tau} v'(s')\right)$$

$$\leq \gamma \max_{a \in \mathcal{A}} \left| \mathcal{P}^\pi_{s's_\tau} \mathcal{P}^a \left(v(s') - v'(s')\right) \right|$$

$$\leq \gamma \left| v(s') - v'(s') \right|$$

$$\leq \gamma \|v - v'\|,$$

*again using the fact that $P$ is non-expansive. Hence we have succeeded in showing that for any $v \in L_2$ we have that*

$$\left\| \mathcal{M}v - \max_{a \in \mathcal{A}} \left[v(\cdot, a) + \gamma \mathcal{P}^a v'\right] \right\| \leq \gamma \|v - v'\|. \tag{18}$$

*Gathering the results of the three cases gives the desired result.*

To prove part ii), we make use of the following result:

**Theorem 5 (Theorem 1, pg 4 in [15])** *Let $\Xi_t(s)$ be a random process that takes values in $\mathbb{R}^n$ and given by the following:*

$$\Xi_{t+1}(s) = (1 - \alpha_t(s)) \Xi_t(s) \alpha_t(s) L_t(s), \tag{19}$$

*then $\Xi_t(s)$ converges to 0 with probability 1 under the following conditions:*

607   *i)* $0 \le \alpha_t \le 1, \sum_t \alpha_t = \infty$ *and* $\sum_t \alpha_t < \infty$

608   *ii)* $\|\mathbb{E}[L_t|\mathcal{F}_t]\| \le \gamma \|\Xi_t\|$, *with* $\gamma < 1$;

609   *iii)* $\mathrm{Var}\,[L_t|\mathcal{F}_t] \le c(1 + \|\Xi_t\|^2)$ *for some* $c > 0$.

610   To prove the result, we show (i) - (iii) hold. Condition (i) holds by choice of learning rate. It therefore
611   remains to prove (ii) - (iii). We first prove (ii). For this, we consider our variant of the Q-learning
612   update rule:

$$Q_{t+1}(s_t, a_t) = Q_t(s_t, a_t)$$
$$+ \alpha_t(s_t, a_t) \left[ \max \left\{ \mathcal{M}Q_t(s_t, a_t), \mathcal{R}(s_t, a_t) + \gamma \max_{a' \in \mathcal{A}} Q_t(s_{t+1}, a') \right\} - Q_t(s_t, a_t) \right].$$

613   After subtracting $Q^\star(s_t, a_t)$ from both sides and some manipulation we obtain that:

$$\Xi_{t+1}(s_t, a_t)$$
$$= (1 - \alpha_t(s_t, a_t))\Xi_t(s_t, a_t)$$
$$+ \alpha_t(s_t, a_t) \left[ \max \left\{ \mathcal{M}Q_t(s_t, a_t), \mathcal{R}(s_t, a_t) + \gamma \max_{a' \in \mathcal{A}} Q_t(s_{t+1}, a') \right\} - Q^\star(s_t, a_t) \right],$$

614   where $\Xi_t(s_t, a_t) := Q_t(s_t, a_t) - Q^\star(s_t, a_t)$.

615   Let us now define by

$$L_t(s_{\tau_k}, a) := \max \left\{ \mathcal{M}Q(s_{\tau_k}, a), \mathcal{R}(s_{\tau_k}, a) + \gamma \max_{a' \in \mathcal{A}} Q(s', a') \right\} - Q^\star(s_t, a).$$

616   Then

$$\Xi_{t+1}(s_t, a_t) = (1 - \alpha_t(s_t, a_t))\Xi_t(s_t, a_t) + \alpha_t(s_t, a_t)\left[L_t(s_{\tau_k}, a)\right]. \tag{20}$$

617   We now observe that

$$\mathbb{E}\left[L_t(s_{\tau_k}, a)|\mathcal{F}_t\right] = \sum_{s' \in \mathcal{S}} P(s'; a, s_{\tau_k}) \max \left\{ \mathcal{M}Q(s_{\tau_k}, a), \mathcal{R}(s_{\tau_k}, a) + \gamma \max_{a' \in \mathcal{A}} Q(s', a') \right\} - Q^\star(s_{\tau_k}, a)$$
$$= T_\phi Q_t(s, a) - Q^\star(s, a). \tag{21}$$

618   Now, using the fixed point property that implies $Q^\star = T_\phi Q^\star$, we find that

$$\mathbb{E}\left[L_t(s_{\tau_k}, a)|\mathcal{F}_t\right] = T_\phi Q_t(s, a) - T_\phi Q^\star(s, a)$$
$$\le \|T_\phi Q_t - T_\phi Q^\star\|$$
$$\le \gamma \|Q_t - Q^\star\|_\infty = \gamma \|\Xi_t\|_\infty. \tag{22}$$

619   using the contraction property of $T$ established in Lemma 1. This proves (ii).

620   We now prove iii), that is

$$\mathrm{Var}\,[L_t|\mathcal{F}_t] \le c(1 + \|\Xi_t\|^2). \tag{23}$$

621   Now by (21) we have that

$$\mathrm{Var}\,[L_t|\mathcal{F}_t] = \mathrm{Var}\left[\max \left\{ \mathcal{M}Q_t(s_t, a_t), \mathcal{R}(s_t, a_t) + \gamma \max_{a' \in \mathcal{A}} Q_t(s_{t+1}, a') \right\} - Q^\star(s, a)\right]$$
$$= \mathbb{E}\left[\left( \max \left\{ \mathcal{M}Q(s_{\tau_k}, a), \mathcal{R}(s_{\tau_k}, a) + \gamma \max_{a' \in \mathcal{A}} Q(s', a') \right\} \right.\right.$$
$$\left.\left. - Q^\star(s_t, a) - (TQ_t(s, a) - Q^\star(s, a)) \right)^2\right]$$
$$= \mathbb{E}\left[\left( \max \left\{ \mathcal{M}Q(s_{\tau_k}, a), \mathcal{R}(s_{\tau_k}, a) + \gamma \max_{a' \in \mathcal{A}} Q(s', a') \right\} - TQ_t(s, a) \right)^2\right]$$
$$= \mathrm{Var}\left[\max \left\{ \mathcal{M}Q_t(s_t, a_t), \mathcal{R}(s_t, a_t) + \gamma \max_{a' \in \mathcal{A}} Q_t(s_{t+1}, a') \right\} - TQ_t(s, a))^2\right]$$
$$\le c(1 + \|\Xi_t\|^2),$$

622   for some $c > 0$ where the last line follows due to the boundedness of $Q$ (which follows from
623   Assumptions 2 and 4). This concludes the proof of the Theorem.

## Proof of Convergence with Linear Function Approximation

First let us recall the statement of the theorem:

**Theorem 3** *LICRA converges to a limit point $r^\star$ which is the unique solution to the equation:*

$$\Pi\mathfrak{F}(\Phi r^\star) = \Phi r^\star, \qquad a.e. \tag{24}$$

*where we recall that for any test function $\psi \in \mathcal{V}$, the operator $\mathfrak{F}$ is defined by $\mathfrak{F}\psi := \mathcal{R} + \gamma P \max\{\mathcal{M}\psi, \psi\}$.*

*Moreover, $r^\star$ satisfies the following:*

$$\|\Phi r^\star - Q^\star\| \le c \|\Pi Q^\star - Q^\star\|. \tag{25}$$

The theorem is proven using a set of results that we now establish. To this end, we first wish to prove the following bound:

**Lemma 4** *For any $Q \in \mathcal{V}$ we have that*

$$\|\mathfrak{F}Q - Q'\| \le \gamma \|Q - Q'\|, \tag{26}$$

*so that the operator $\mathfrak{F}$ is a contraction.*

**Proof 4** *Recall, for any test function $\psi$, a projection operator $\Pi$ acting $\psi$ is defined by the following*

$$\Pi\psi := \operatorname*{arg\,min}_{\bar{\psi} \in \{\Phi r | r \in \mathbb{R}^p\}} \|\bar{\psi} - \psi\|.$$

*Now, we first note that in the proof of Lemma 1, we deduced that for any $\psi \in L_2$ we have that*

$$\left\| \mathcal{M}\psi - \left[ \mathcal{R}(\cdot, a) + \gamma \max_{a \in \mathcal{A}} \mathcal{P}^a \psi' \right] \right\| \le \gamma \|\psi - \psi'\|,$$

*(c.f. Lemma 1).*

*Setting $\psi = Q$ and $\psi' = Q'$ it can be straightforwardly deduced that for any $Q, \hat{Q} \in L_2$: $\left\| \mathcal{M}Q - \hat{Q} \right\| \le \gamma \left\| Q - \hat{Q} \right\|$. Hence, using the contraction property of $\mathcal{M}$, we readily deduce the following bound:*

$$\max\left\{ \left\| \mathcal{M}Q - \hat{Q} \right\|, \left\| \mathcal{M}Q - \mathcal{M}\hat{Q} \right\| \right\} \le \gamma \left\| Q - \hat{Q} \right\|, \tag{27}$$

*We now observe that $\mathfrak{F}$ is a contraction. Indeed, since for any $Q, Q' \in L_2$ we have that:*

$$\begin{aligned}
\|\mathfrak{F}Q - \mathfrak{F}Q'\| &= \|\mathcal{R} + \gamma P \max\{\mathcal{M}Q, Q\} - (\mathcal{R} + \gamma P \max\{\mathcal{M}Q', Q'\})\| \\
&= \gamma \|P \max\{\mathcal{M}Q, Q\} - P \max\{\mathcal{M}Q', Q'\}\| \\
&\le \gamma \|\max\{\mathcal{M}Q, Q\} - \max\{\mathcal{M}Q', Q'\}\| \\
&\le \gamma \|\max\{\mathcal{M}Q - \mathcal{M}Q', Q - \mathcal{M}Q', \mathcal{M}Q - Q', Q - Q'\}\| \\
&\le \gamma \max\{\|\mathcal{M}Q - \mathcal{M}Q'\|, \|Q - \mathcal{M}Q'\|, \|\mathcal{M}Q - Q'\|, \|Q - Q'\|\} \\
&= \gamma \|Q - Q'\|,
\end{aligned}$$

*using (27) and again using the non-expansiveness of $P$.*

We next show that the following two bounds hold:

**Lemma 5** *For any $Q \in \mathcal{V}$ we have that*

*i)* $\qquad \|\Pi\mathfrak{F}Q - \Pi\mathfrak{F}\bar{Q}\| \le \gamma \|Q - \bar{Q}\|,$

*ii)* $\qquad \|\Phi r^\star - Q^\star\| \le \frac{1}{\sqrt{1-\gamma^2}} \|\Pi Q^\star - Q^\star\|.$

25

**Proof 5** *The first result is straightforward since as $\Pi$ is a projection it is non-expansive and hence:*

$$\left\|\Pi\mathfrak{F}Q - \Pi\mathfrak{F}\bar{Q}\right\| \leq \left\|\mathfrak{F}Q - \mathfrak{F}\bar{Q}\right\| \leq \gamma \left\|Q - \bar{Q}\right\|,$$

*using the contraction property of $\mathfrak{F}$. This proves i). For ii), we note that by the orthogonality property of projections we have that $\langle \Phi r^\star - \Pi Q^\star, \Phi r^\star - \Pi Q^\star \rangle$, hence we observe that:*

$$\begin{aligned}
\left\|\Phi r^\star - Q^\star\right\|^2 &= \left\|\Phi r^\star - \Pi Q^\star\right\|^2 + \left\|\Phi r^\star - \Pi Q^\star\right\|^2 \\
&= \left\|\Pi\mathfrak{F}\Phi r^\star - \Pi Q^\star\right\|^2 + \left\|\Phi r^\star - \Pi Q^\star\right\|^2 \\
&\leq \left\|\mathfrak{F}\Phi r^\star - Q^\star\right\|^2 + \left\|\Phi r^\star - \Pi Q^\star\right\|^2 \\
&= \left\|\mathfrak{F}\Phi r^\star - \mathfrak{F}Q^\star\right\|^2 + \left\|\Phi r^\star - \Pi Q^\star\right\|^2 \\
&\leq \gamma^2 \left\|\Phi r^\star - Q^\star\right\|^2 + \left\|\Phi r^\star - \Pi Q^\star\right\|^2,
\end{aligned}$$

*after which we readily deduce the desired result.*

**Lemma 6** *Define the operator $H$ by the following:* $HQ(s,a) =$
$$\begin{cases} \mathcal{M}Q(s,a), & \text{if } \mathcal{M}Q(s,a) > \Phi r^\star, \\ Q(s,a), & \text{otherwise,} \end{cases}$$
*where we define $\tilde{\mathfrak{F}}$ by: $\tilde{\mathfrak{F}}Q := \mathcal{R} + \gamma PHQ$.*

*For any $Q, \bar{Q} \in L_2$ we have that*

$$\left\|\tilde{\mathfrak{F}}Q - \tilde{\mathfrak{F}}\bar{Q}\right\| \leq \gamma \left\|Q - \bar{Q}\right\| \tag{28}$$

*and hence $\tilde{\mathfrak{F}}$ is a contraction mapping.*

**Proof 6** *Using (27), we now observe that*

$$\begin{aligned}
\left\|\tilde{\mathfrak{F}}Q - \tilde{\mathfrak{F}}\bar{Q}\right\| &= \left\|\mathcal{R} + \gamma PHQ - \left(\mathcal{R} + \gamma PH\bar{Q}\right)\right\| \\
&\leq \gamma \left\|HQ - H\bar{Q}\right\| \\
&\leq \gamma \left\|\max\left\{\mathcal{M}Q - \mathcal{M}\bar{Q}, Q - \bar{Q}, \mathcal{M}Q - \bar{Q}, \mathcal{M}\bar{Q} - Q\right\}\right\| \\
&\leq \gamma \max\left\{\left\|\mathcal{M}Q - \mathcal{M}\bar{Q}\right\|, \left\|Q - \bar{Q}\right\|, \left\|\mathcal{M}Q - \bar{Q}\right\|, \left\|\mathcal{M}\bar{Q} - Q\right\|\right\} \\
&\leq \gamma \max\left\{\gamma \left\|Q - \bar{Q}\right\|, \left\|Q - \bar{Q}\right\|, \left\|\mathcal{M}Q - \bar{Q}\right\|, \left\|\mathcal{M}\bar{Q} - Q\right\|\right\} \\
&= \gamma \left\|Q - \bar{Q}\right\|,
\end{aligned}$$

*again using the non-expansive property of $P$.*

**Lemma 7** *Define by $\tilde{Q} := \mathcal{R} + \gamma P v^{\tilde{\pi}}$ where*

$$v^{\tilde{\pi}}(s) := \mathcal{R}(s_{\tau_k}, a) + \gamma \max_{a \in \mathcal{A}} \sum_{s' \in \mathcal{S}} P(s'; a, s_{\tau_k}) \Phi r^\star(s'), \tag{29}$$

*then $\tilde{Q}$ is a fixed point of $\tilde{\mathfrak{F}}\tilde{Q}$, that is $\tilde{\mathfrak{F}}\tilde{Q} = \tilde{Q}$.*

**Proof 7** *We begin by observing that*

$$\begin{aligned}
H\tilde{Q}(s,a) &= H\left(\mathcal{R}(s,\cdot) + \gamma P v^{\tilde{\pi}}\right) \\
&= \begin{cases} \mathcal{M}Q(s,a), & \text{if } \mathcal{M}Q(s,a) > \Phi r^\star, \\ Q(s,a), & \text{otherwise,} \end{cases} \\
&= \begin{cases} \mathcal{M}Q(s,a), & \text{if } \mathcal{M}Q(s,a) > \Phi r^\star, \\ \mathcal{R}(s,\cdot) + \gamma P v^{\tilde{\pi}}, & \text{otherwise,} \end{cases} \\
&= v^{\tilde{\pi}}(s).
\end{aligned}$$

*Hence,*

$$\tilde{\mathfrak{F}}\tilde{Q} = \mathcal{R} + \gamma PH\tilde{Q} = \mathcal{R} + \gamma P v^{\tilde{\pi}} = \tilde{Q}. \tag{30}$$

*which proves the result.*

**Lemma 8** *The following bound holds:*

$$\mathbb{E}\left[v^{\hat{\pi}}(s_0)\right] - \mathbb{E}\left[v^{\tilde{\pi}}(s_0)\right] \le 2\left[(1-\gamma)\sqrt{(1-\gamma^2)}\right]^{-1}\|\Pi Q^\star - Q^\star\|. \tag{31}$$

**Proof 8** *By definitions of $v^{\hat{\pi}}$ and $v^{\tilde{\pi}}$ (c.f (29)) and using Jensen's inequality and the stationarity property we have that,*

$$
\begin{aligned}
\mathbb{E}\left[v^{\hat{\pi}}(s_0)\right] - \mathbb{E}\left[v^{\tilde{\pi}}(s_0)\right] &= \mathbb{E}\left[Pv^{\hat{\pi}}(s_0)\right] - \mathbb{E}\left[Pv^{\tilde{\pi}}(s_0)\right] \\
&\le \left|\mathbb{E}\left[Pv^{\hat{\pi}}(s_0)\right] - \mathbb{E}\left[Pv^{\tilde{\pi}}(s_0)\right]\right| \\
&\le \left\|Pv^{\hat{\pi}} - Pv^{\tilde{\pi}}\right\|.
\end{aligned} \tag{32}
$$

*Now recall that $\tilde{Q} := \mathcal{R} + \gamma Pv^{\tilde{\pi}}$ and $Q^\star := \mathcal{R} + \gamma Pv^{\pi^\star}$, using these expressions in (32) we find that*

$$\mathbb{E}\left[v^{\hat{\pi}}(s_0)\right] - \mathbb{E}\left[v^{\tilde{\pi}}(s_0)\right] \le \frac{1}{\gamma}\left\|\tilde{Q} - Q^\star\right\|.$$

*Moreover, by the triangle inequality and using the fact that $\mathfrak{F}(\Phi r^\star) = \tilde{\mathfrak{F}}(\Phi r^\star)$ and that $\mathfrak{F}Q^\star = Q^\star$ and $\mathfrak{F}\tilde{Q} = \tilde{Q}$ (c.f. (31)) we have that*

$$
\begin{aligned}
\left\|\tilde{Q} - Q^\star\right\| &\le \left\|\tilde{Q} - \mathfrak{F}(\Phi r^\star)\right\| + \left\|Q^\star - \tilde{\mathfrak{F}}(\Phi r^\star)\right\| \\
&\le \gamma\left\|\tilde{Q} - \Phi r^\star\right\| + \gamma\left\|Q^\star - \Phi r^\star\right\| \\
&\le 2\gamma\left\|\tilde{Q} - \Phi r^\star\right\| + \gamma\left\|Q^\star - \tilde{Q}\right\|,
\end{aligned}
$$

*which gives the following bound:*

$$\left\|\tilde{Q} - Q^\star\right\| \le 2\left(1-\gamma\right)^{-1}\left\|\tilde{Q} - \Phi r^\star\right\|,$$

*from which, using Lemma 5, we deduce that $\left\|\tilde{Q} - Q^\star\right\| \le 2\left[(1-\gamma)\sqrt{(1-\gamma^2)}\right]^{-1}\left\|\tilde{Q} - \Phi r^\star\right\|$, after which by (33), we finally obtain*

$$\mathbb{E}\left[v^{\hat{\pi}}(s_0)\right] - \mathbb{E}\left[v^{\tilde{\pi}}(s_0)\right] \le 2\left[(1-\gamma)\sqrt{(1-\gamma^2)}\right]^{-1}\left\|\tilde{Q} - \Phi r^\star\right\|,$$

*as required.*

Let us rewrite the update in the following way:

$$r_{t+1} = r_t + \gamma_t \Xi(w_t, r_t),$$

where the function $\Xi : \mathbb{R}^{2d} \times \mathbb{R}^p \to \mathbb{R}^p$ is given by:

$$\Xi(w, r) := \phi(s)\left(\mathcal{R}(s, \cdot) + \gamma \max\left\{(\Phi r)(s'), \mathcal{M}(\Phi r)(s')\right\} - (\Phi r)(s)\right),$$

for any $w \equiv (s, s') \in \mathcal{S}^2$ and for any $r \in \mathbb{R}^p$. Let us also define the function $\boldsymbol{\Xi} : \mathbb{R}^p \to \mathbb{R}^p$ by the following:

$$\boldsymbol{\Xi}(r) := \mathbb{E}_{w_0 \sim (\mathbb{P}, \mathbb{P})}\left[\Xi(w_0, r)\right]; w_0 := (s_0, z_1).$$

**Lemma 9** *The following statements hold for all $z \in \{0, 1\} \times \mathcal{S}$:*

*i) $(r - r^\star)\boldsymbol{\Xi}_k(r) < 0, \qquad \forall r \neq r^\star,$*

*ii) $\boldsymbol{\Xi}_k(r^\star) = 0.$*

**Proof 9** *To prove the statement, we first note that each component of $\boldsymbol{\Xi}_k(r)$ admits a representation as an inner product, indeed:*

$$
\begin{aligned}
\boldsymbol{\Xi}_k(r) &= \mathbb{E}\left[\phi_k(s_0)(\mathcal{R}(s_0, a_0) + \gamma \max\left\{\Phi r(s_1), \mathcal{M}\Phi(s_1)\right\} - (\Phi r)(s_0)\right] \\
&= \mathbb{E}\left[\phi_k(s_0)(\mathcal{R}(s_0, a_0) + \gamma\mathbb{E}\left[\max\left\{\Phi r(s_1), \mathcal{M}\Phi(s_1)\right\}|z_0\right] - (\Phi r)(s_0)\right] \\
&= \mathbb{E}\left[\phi_k(s_0)(\mathcal{R}(s_0, a_0) + \gamma P \max\left\{(\Phi r, \mathcal{M}\Phi)\right\}(s_0) - (\Phi r)(s_0)\right] \\
&= \langle \phi_k, \mathfrak{F}\Phi r - \Phi r\rangle,
\end{aligned}
$$

683   *using the iterated law of expectations and the definitions of $P$ and $\mathfrak{F}$.*

684   *We now are in position to prove i). Indeed, we now observe the following:*

$$
\begin{aligned}
(r - r^\star)\,\boldsymbol{\Xi}_k(r) &= \sum_{l=1} (r(l) - r^\star(l))\,\langle \phi_l, \mathfrak{F}\Phi r - \Phi r \rangle \\
&= \langle \Phi r - \Phi r^\star, \mathfrak{F}\Phi r - \Phi r \rangle \\
&= \langle \Phi r - \Phi r^\star, (\mathbf{1} - \Pi)\mathfrak{F}\Phi r + \Pi\mathfrak{F}\Phi r - \Phi r \rangle \\
&= \langle \Phi r - \Phi r^\star, \Pi\mathfrak{F}\Phi r - \Phi r \rangle,
\end{aligned}
$$

685   *where in the last step we used the orthogonality of $(\mathbf{1} - \Pi)$. We now recall that $\Pi\mathfrak{F}\Phi r^\star = \Phi r^\star$*
686   *since $\Phi r^\star$ is a fixed point of $\Pi\mathfrak{F}$. Additionally, using Lemma 5 we observe that $\|\Pi\mathfrak{F}\Phi r - \Phi r^\star\| \le$*
687   $\gamma\|\Phi r - \Phi r^\star\|$. *With this we now find that*

$$
\begin{aligned}
&\langle \Phi r - \Phi r^\star, \Pi\mathfrak{F}\Phi r - \Phi r \rangle \\
&= \langle \Phi r - \Phi r^\star, (\Pi\mathfrak{F}\Phi r - \Phi r^\star) + \Phi r^\star - \Phi r \rangle \\
&\le \|\Phi r - \Phi r^\star\|\,\|\Pi\mathfrak{F}\Phi r - \Phi r^\star\| - \|\Phi r^\star - \Phi r\|^2 \\
&\le (\gamma - 1)\,\|\Phi r^\star - \Phi r\|^2,
\end{aligned}
$$

688   *which is negative since $\gamma < 1$ which completes the proof of part i).*

689   *The proof of part ii) is straightforward since we readily observe that*

$$
\boldsymbol{\Xi}_k(r^\star) = \langle \phi_l, \mathfrak{F}\Phi r^\star - \Phi r \rangle = \langle \phi_l, \Pi\mathfrak{F}\Phi r^\star - \Phi r \rangle = 0,
$$

690   *as required and from which we deduce the result.*

691   To prove the theorem, we make use of a special case of the following result:

692   **Theorem 6 (Th. 17, p. 239 in [5])** *Consider a stochastic process $r_t : \mathbb{R} \times \{\infty\} \times \Omega \to \mathbb{R}^k$ which*
693   *takes an initial value $r_0$ and evolves according to the following:*

$$
r_{t+1} = r_t + \alpha\Xi(s_t, r_t), \tag{33}
$$

694   *for some function $s : \mathbb{R}^{2d} \times \mathbb{R}^k \to \mathbb{R}^k$ and where the following statements hold:*

695     *1. $\{s_t | t = 0, 1, \ldots\}$ is a stationary, ergodic Markov process taking values in $\mathbb{R}^{2d}$*

696     *2. For any positive scalar $q$, there exists a scalar $\mu_q$ such that $\mathbb{E}\left[1 + \|s_t\|^q \,|\, s \equiv s_0\right] \le$*
697       $\mu_q\left(1 + \|s\|^q\right)$

698     *3. The step size sequence satisfies the Robbins-Monro conditions, that is $\sum_{t=0}^{\infty} \alpha_t = \infty$ and*
699       $\sum_{t=0}^{\infty} \alpha_t^2 < \infty$

700     *4. There exists scalars $c$ and $q$ such that $\|\Xi(w, r)\| \le c\left(1 + \|w\|^q\right)\left(1 + \|r\|\right)$*

701     *5. There exists scalars $c$ and $q$ such that $\sum_{t=0}^{\infty} \|\mathbb{E}\left[\Xi(w_t, r)\,|\,z_0 \equiv z\right] - \mathbb{E}\left[\Xi(w_0, r)\right]\| \le$*
702       $c\left(1 + \|w\|^q\right)\left(1 + \|r\|\right)$

703     *6. There exists a scalar $c > 0$ such that $\|\mathbb{E}[\Xi(w_0, r)] - \mathbb{E}[\Xi(w_0, \bar{r})]\| \le c\|r - \bar{r}\|$*

704     *7. There exists scalars $c > 0$ and $q > 0$ such that*
705       $\sum_{t=0}^{\infty} \|\mathbb{E}\left[\Xi(w_t, r)\,|\,w_0 \equiv w\right] - \mathbb{E}\left[\Xi(w_0, \bar{r})\right]\| \le c\|r - \bar{r}\|\left(1 + \|w\|^q\right)$

706     *8. There exists some $r^\star \in \mathbb{R}^k$ such that $\boldsymbol{\Xi}(r)(r - r^\star) < 0$ for all $r \ne r^\star$ and $\bar{s}(r^\star) = 0$.*

707   *Then $r_t$ converges to $r^\star$ almost surely.*

708   In order to apply the Theorem 6, we show that conditions 1 - 7 are satisfied.

709   **Proof 10** *Conditions 1-2 are true by assumption while condition 3 can be made true by choice of the*
710   *learning rates. Therefore it remains to verify conditions 4-7 are met.*

711 *To prove 4, we observe that*

$$\|\Xi(w,r)\| = \|\phi(s)\left(\mathcal{R}(s,\cdot) + \gamma \max\left\{(\Phi r)(s'), \mathcal{M}\Phi(s')\right\} - (\Phi r)(s)\right)\|$$
$$\leq \|\phi(s)\| \|\mathcal{R}(s,\cdot) + \gamma\left(\|\phi(s')\| \|r\| + \mathcal{M}\Phi(s')\right)\| + \|\phi(s)\| \|r\|$$
$$\leq \|\phi(s)\| \left(\|\mathcal{R}(s,\cdot)\| + \gamma\|\mathcal{M}\Phi(s')\|\right) + \|\phi(s)\| \left(\gamma \|\phi(s')\| + \|\phi(s)\|\right) \|r\|.$$

712 *Now using the definition of $\mathcal{M}$, we readily observe that $\|\mathcal{M}\Phi(s')\| \leq \|\mathcal{R}\| + \gamma\|\mathcal{P}^{\pi}_{s's_t}\Phi\| \leq \|\mathcal{R}\| +$*
713 *$\gamma\|\Phi\|$ using the non-expansiveness of $P$.*

714 *Hence, we lastly deduce that*

$$\|\Xi(w,r)\| \leq \|\phi(s)\| \left(\|\mathcal{R}(s,\cdot)\| + \gamma\|\mathcal{M}\Phi(s')\|\right) + \|\phi(s)\| \left(\gamma \|\phi(s')\| + \|\phi(s)\|\right) \|r\|$$
$$\leq \|\phi(s)\| \left(\|\mathcal{R}(s,\cdot)\| + \gamma\|\mathcal{R}\| + \gamma\|\phi\|\right) + \|\phi(s)\| \left(\gamma \|\phi(s')\| + \|\phi(s)\|\right) \|r\|,$$

715 *we then easily deduce the result using the boundedness of $\phi$ and $\mathcal{R}$.*

716 *Now we observe the following Lipschitz condition on $\Xi$:*

$$\|\Xi(w,r) - \Xi(w,\bar{r})\|$$
$$= \|\phi(s)\left(\gamma \max\left\{(\Phi r)(s'), \mathcal{M}\Phi(s')\right\} - \gamma \max\left\{(\Phi\bar{r})(s'), \mathcal{M}\Phi(s')\right\}\right) - \left((\Phi r)(s) - \Phi\bar{r}(s)\right)\|$$
$$\leq \gamma \|\phi(s)\| \|\max\left\{\phi'(s')r, \mathcal{M}\Phi'(s')\right\} - \max\left\{(\phi'(s')\bar{r}), \mathcal{M}\Phi'(s')\right\}\| + \|\phi(s)\| \|\phi'(s)r - \phi(s)\bar{r}\|$$
$$\leq \gamma \|\phi(s)\| \|\phi'(s')r - \phi'(s')\bar{r}\| + \|\phi(s)\| \|\phi'(s)r - \phi(s)\bar{r}\|$$
$$\leq \|\phi(s)\| \left(\|\phi(s)\| + \gamma \|\phi(s)\| \|\phi'(s') - \phi'(s')\|\right) \|r - \bar{r}\|$$
$$\leq c \|r - \bar{r}\|,$$

717 *using Cauchy-Schwarz inequality and that for any scalars $a, b, c$ we have that*
718 *$|\max\{a, b\} - \max\{b, c\}| \leq |a - c|$.*

719 *Using Assumptions 3 and 4, we therefore deduce that*

$$\sum_{t=0}^{\infty} \|\mathbb{E}\left[\Xi(w,r) - \Xi(w,\bar{r})|w_0 = w\right] - \mathbb{E}\left[\Xi(w_0, r) - \Xi(w_0, \bar{r})\right]\| \leq c \|r - \bar{r}\| \left(1 + \|w\|^l\right). \quad (34)$$

720 *Part 2 is assured by Lemma 5 while Part 4 is assured by Lemma 8 and lastly Part 8 is assured by*
721 *Lemma 9. This result completes the proof of Theorem 1.*

## Proof of Proposition 1

723 **Proof 11** *First let us recall that the intervention time $\tau_k$ is defined recursively $\tau_k = \inf\{t >$*
724 *$\tau_{k-1}|s_t \in A, \tau_k \in \mathcal{F}_t\}$ where $A = \{s \in \mathcal{S}, g(s_t) = 1\}$. The proof is given by establishing a*
725 *contradiction. Therefore suppose that $\mathcal{M}\psi(s_{\tau_k}) \leq \psi(s_{\tau_k})$ and suppose that the intervention time*
726 *$\tau'_1 > \tau_1$ is an optimal intervention time. Construct the $\pi' \in \Pi$ and $\tilde{\pi} \in \Pi$ policy switching times*
727 *by $(\tau'_0, \tau'_1, \ldots,)$ and $(\tau'_0, \tau_1, \ldots)$ respectively. Define by $l = \inf\{t > 0; \mathcal{M}\psi(s_t) = \psi(s_t)\}$ and*
728 *$m = \sup\{t; t < \tau'_1\}$. By construction we have that*

$$v^{\pi'}(s)$$
$$= \mathbb{E}\left[\mathcal{R}(s_0, a_0) + \mathbb{E}\left[\ldots + \gamma^{l-1}\mathbb{E}\left[\mathcal{R}(s_{\tau_1-1}, a_{\tau_1-1}) + \ldots + \gamma^{m-l-1}\mathbb{E}\left[\mathcal{R}(s_{\tau'_1-1}, a_{\tau'_1-1}) + \gamma\mathcal{M}^{\pi^1, \pi'}v^{\pi'}(s', I(\tau'_1))\right]\right]\right]\right]$$
$$< \mathbb{E}\left[\mathcal{R}(s_0, a_0) + \mathbb{E}\left[\ldots + \gamma^{l-1}\mathbb{E}\left[\mathcal{R}(s_{\tau_1-1}, a_{\tau_1-1}) + \gamma\mathcal{M}^{\tilde{\pi}}v^{\pi'}(s_{\tau_1})\right]\right]\right]$$

729 *We now use the following observation*

$$\mathbb{E}\left[\mathcal{R}(s_{\tau_1-1}, a_{\tau_1-1}) + \gamma\mathcal{M}^{\tilde{\pi}}v^{\pi'}(s_{\tau_1})\right] \quad (35)$$
$$\leq \max\left\{\mathcal{M}^{\tilde{\pi}}v^{\pi'}(s_{\tau_1}), \max_{a_{\tau_1} \in \mathcal{A}}\left[\mathcal{R}(s_{\tau_k}, a_{\tau_k}) + \gamma \sum_{s' \in \mathcal{S}} P(s'; a_{\tau_1}, s_{\tau_1})v^{\pi}(s')\right]\right\}. \quad (36)$$

 *Using this we deduce that*

$$v^{\pi'}(s) \le \mathbb{E}\left[\mathcal{R}(s_0, a_0) + \mathbb{E}\left[\dots\right.\right.$$

$$\left.\left. + \gamma^{l-1}\mathbb{E}\left[\mathcal{R}(s_{\tau_1-1}, a_{\tau_1-1}) + \gamma \max\left\{\mathcal{M}^{\tilde{\pi}} v^{\pi'}(s_{\tau_1}), \max_{a_{\tau_1} \in \mathcal{A}}\left[\mathcal{R}(s_{\tau_k}, a_{\tau_k}) + \gamma \sum_{s' \in \mathcal{S}} P(s'; a_{\tau_1}, s_{\tau_1}) v^{\pi}(s')\right]\right\}\right]\right]\right]$$

$$= \mathbb{E}\left[\mathcal{R}(s_0, a_0) + \mathbb{E}\left[\dots + \gamma^{l-1}\mathbb{E}\left[\mathcal{R}(s_{\tau_1-1}, a_{\tau_1-1}) + \gamma\left[T v^{\tilde{\pi}}\right](s_{\tau_1})\right]\right]\right] = v^{\tilde{\pi}}(s)),$$

731   *where the first inequality is true by assumption on $\mathcal{M}$. This is a contradiction since $\pi'$ is an optimal*
732   *policy for Player 2. Using analogous reasoning, we deduce the same result for $\tau'_k < \tau_k$ after which*
733   *deduce the result. Moreover, by invoking the same reasoning, we can conclude that it must be the*
734   *case that $(\tau_0, \tau_1, \dots, \tau_{k-1}, \tau_k, \tau_{k+1}, \dots,)$ are the optimal switching times.*