# OpenReview forum: "Timing is Everything: Learning to Act Selectively with Costly Actions and Budgetary Constraints"
_NeurIPS.cc/2022/Conference — NeurIPS 2022 Submitted_

### Official Review · Reviewer_UN8M · 2022-07-02

**Rating:** 5
**Confidence:** 3
**Soundness:** 2 fair
**Presentation:** 2 fair
**Contribution:** 2 fair

**Summary:**

This paper studies a class of sequential decision making problems, in which taking an action can incur a large cost. To address these problems, they proposed a method named Learnable Impulse Control Reinforcement Algorithm (LICRA), which learns both when to act and which action to take. For the proposed algorithm, they introduced a new Bellman operator and established convergence results. Then they generalized LICRA with state augmentation to solve the RL problem with a budget. Finally, they tested LICRL on three tasks.

**Questions:**

1. The operator of $\mathcal{M}^{\pi, g}$. This operator should be applied on **any** function in $S \times A \rightarrow \mathbb{R}$. Why use the notation $Q^{\pi, g}$, which depends on $(\pi, g)$? Besides, the function $\mathcal{M}^{\pi, g} Q^{\pi, g}$ should be dependent on $Q^{\pi, g}$. However, according to the definition given in line 186, $\mathcal{M}^{\pi, g} Q^{\pi, g}$ is independent of $Q^{\pi, g}$. Besides, in line 191, what does the notation $\mathcal{M} v^{\pi, g}$ mean?
2. The operator T in line 195. The operator T does not depend on any policy $(\pi, g)$. However, in its definition, the terms $\mathcal{M}^{\pi, g}$, $Q^{\pi, g}$ and $v^{\pi, g}$ all depend on some policy $(\pi, g)$,


**Ethics Review Area:**

["I don’t know"]

**Limitations:**

The authors do not mention any limitations or potential negative societal impact of their work.

**Strengths And Weaknesses:**

Strengths:
1. They proposed a reasonable LICRA algorithm that utilizes two separate policies to learn when to act and which action to take. It was shown that LICRA can also solve the RL problem with a budget.


Weakness:
1. The proposed method LICRA is not well-motivated and it is unclear why we need to learn two separate policies. As mentioned by the authors, to address RL problems in which taking an action can incur a cost, we can simply augment the action space with a null action and apply classical RL algorithms. Compared with this simple method, LICRA has an identical search space (action space) but requires twice the computational cost for learning the additional actor and critic. Besides, constrained RL approaches can also be applied in this setting. The authors do not show the advantages of LICRA over the methods mentioned before.
2. The empirical evaluations are weak. First, the algorithms for comparison are too few. The constrained RL algorithms should be involved for comparison. Second, the improvements of LICRA are not clear on Merton’s Portfolio Problem and Lunar Lander task. Third, the introductions of the baseline algorithms in Figure 2 and Figure 3 are missing.
3. The theoretical results and notations in Sec 5 are misleading. See the question part for details. I encourage the authors to re-write this part to improve the clarity.
4. This paper studies RL problems where taking an action can incur a large cost. A closely related work [1] is missing. [1] studies the RL problems where the amounts of some specific actions are limited, which is similar to the problems studied in this paper.

References:
[1]. Ji-Cheng, Pang, et al. "Sparsity Prior Regularized Q-learning for Sparse Action Tasks.”arXiv preprint, arXiv: 2105.08666, (2021).

---

> ### Author Response · Authors · 2022-08-02
> **Response to the reviewer**
>
> We thank the reviewer for their reading and comments. We hope that our response will enable the reviewer to appreciate the clarity of the paper and motivation as the other reviewers have.

---

> ### Author Response · Authors · 2022-08-02
> **Response to the reviewer UN8M**
>
> > "The proposed method LICRA is not well-motivated and it is unclear why we need to learn two separate policies."
>
> #### Answer:
>
> In Sec. 4, we expended considerable effort to explain the benefits of our method and justifying its structure. We refer the reviewer to lines 144- 172 where this is explained in detail. We have now also added several analyses to explain our claims. Specifically, we validated our claim that LICRA's structure leads to large efficiency gains relative to current state-of-the-art RL methods when the agent must be selective about where it acts. Our ablation study (Ablation study 1) reveals that LICRA's dual structure enables it to learn to be highly selective about where to act and where it should learn the corresponding optimal magnitudes for the actions. The reviewer is correct in saying that LICRA requires more computation - as we demonstrate in our numerous analyses, this enables RL to solve problems of this kind and achieve optimal performance unlike current methods which in some cases fail to solve the task e.g. the Drive problem (c.f. Fig. 2) where LICRA achieves maximal performance of a score of $2.5$ and the baselines achieve a score of $0$ indicating their failure to solve the task.
>
>
> > "The empirical evaluations are weak...the algorithms for comparison are too few."
>
> #### Answer:
>
> We thank the reviewer for their comment. We emphasise the LICRA is a plug \& play method which enables existing RL methods to be easily integrated into LICRA. Therefore the most relevant comparison is against the base learner. For the sake of completeness, we have now added state-of-the-art RL methods from the constrained RL baselines (Lagrangian based) so that each experiment as baselines leading to a wide coverage of current methods. We show that in all cases the LICRA variants outperform the baselines which in some cases fail to even solve the task (e.g. Drive drive problem in Fig. 2 and Fig. 5) or are much slower in obtaining the optimal policy e.g. the Lunar Lander probelm in Fig. 3 and the Merton portfolio problem in Fig. .
>
> We are happy to add additional baselines and experiment that reviewer deems necessary for acceptance.
>
> > The constrained RL algorithms should be involved for comparison.
>
> #### Answer:
>
> We have added constrained RL baselines (Lagrangian based) when approriate. We are happy to add further baselines to our experiments.
>
> > The improvements of LICRA are not clear on Merton’s Portfolio Problem and Lunar Lander task.
>
> #### Answer:
>
> As pointed our by reviewer wRMk, the plots may not do the superior performance LICRA achieves justice.
>
> For the Lunar Lander experiment and Drive Environment, we have now added the following final scores to the captions that clearly indicate LICRA's superior empirical performance relative to the base learner.
> * LunarLander(final scores):
> LICRA_SAC(98.03), SAC(61.20), LICRA_PPO(58.84), PPO(49.77).
> * Drive Env(final scores):
> LICRA_SAC(1.75), SAC(0.37), LICRA_PPO(2.63), PPO(0.24)
>
> We chose the Merton problem due to its simplicity ($1$ dimensional state space and action space) and widespread application. Even in this setting, we show that PPO integrated LICRA (LICRA_PPO) solves the problem whereas PPO fails to reach the optimal solution after 500k training steps. Additionally, even though SAC eventually reaches similar performance as LICRA_SAC, SAC alone takes around 250k more training steps to do so in keeping with our claim that LICRA markedly improves the efficiency for tackling problems of this kind.
>
> Each of these evaluations clearly indicate LICRA's superior performance across a range of settings.

---

> > ### Author Response · Authors · 2022-08-03
> > **Response to the reviewer UN8M continued....**
> >
> >
> > >The introductions of the baseline algorithms in Figure 2 and Figure 3 are missing.
> >
> > #### Answer:
> >
> > Thanks, we will add a brief note about the baselines in our updated version.
> >
> > > Clarity and notations in Sec 5.
> >
> > #### Answer:
> >
> > We have adjusted the notation of Sec. 5 to make it more clear and avoid any ambiguity. Please note that none of the analyses or conlusions are affected by these adjustments.
> >
> > > The reference [1] "Sparsity Prior Regularized Q-learning for Sparse Action Tasks.”arXiv preprint, arXiv: 2105.08666, (2021).
> >
> > Thank you for bringing this paper up. It's a pity that we didn't see this paper as it only appeared on arxiv and openreview and hasn't been published in a conference to our best knowledge. It is indeed a related work which highlights the need for treatment of problems of this kind. There are however fundamental differences between the problem settings and approaches. Unlike the approach taken in [1], the problem setting we consider is one in which the agent faces a cost for each action - the produces a need for the agent to be selective about where it performs actions (but does not necessarily constrain the magnitude or choice of those actions). In [1], the setting is one in which the "opportunities of taking sparse action are limited". The authors in [1] also do not give a theoretical treatment of ``sparse actions'' and do not suggest a formalism for tackling this setting but rather propose a pure algorithmic development  (but with good results). We feel that the lack of formalisation prevents consideration of their approach to continuous sparse action spaces and requires  restricting their consideration to discrete action spaces. Our formalisation as impulse control allows for a general approach.
> >
> >
> > > The definition of the operator ${\cal M}^{\pi, g}$.
> >
> > Thanks for the comment. The dependence of $\mathcal{M} Q^{\pi,\mathfrak{g}}$ on $Q^{\pi,\mathfrak{g}}$ is apparent in the value function $v^{\pi,\mathfrak{g}}$ (which has a dependence of $Q^{\pi,\mathfrak{g}}$) which appears in the definition of $\cal M$. Given the definition of the action-value function $Q$; ${\cal M}^{\pi,\mathfrak{g}}$ acting on $Q^{\pi',\mathfrak{g}'}$ represents the expected return when an action intervention is immediately sampled from $\pi$ (and the agent incurs an action cost $c$) and thereafter the policy $\pi'$ is executed.
> >
> > We have updated the definition to the following to avoid this confusion: ""Given a function $Q^{\pi,\mathfrak{g}}:\mathcal{S}\times\mathcal{A}\to\mathbb{R},\;\forall\pi,\pi'\in\Pi$ and $\mathfrak{g},\mathfrak{g}'$, $\forall s_{\tau_k}\in\mathcal{S}$, we define the intervention operator $$\mathcal{M}^{\pi,\mathfrak{g}}$$ by $$\mathcal{M}^{\pi,\mathfrak{g}}Q^{\pi',\mathfrak{g}'}(s_{\tau_k},a_{\tau_k})\\:=\mathcal{R}(s_{\tau_k},a_{\tau_k})-c(s_{\tau_k},a_{\tau_k})+\gamma\sum_{s'\in\mathcal{S}}P(s';a_{\tau_k},s)v^{\pi',\mathfrak{g}'}(s')\Big|a_{\tau_k}\sim\pi$$
> >
> > > The definition of the Bellman operator $T$ and dependence on any policy $(\pi,g)$.
> >
> >
> > The operator $T$ can act on functions, which depend on  $(\pi,g)$, but they can also act on functions that do not. In our case, they act on the functions that depend on $(\pi,g)$.

---

> > > ### Comment · Reviewer_UN8M · 2022-08-06
> > > **Response to the authors**
> > >
> > > Thank the authors for the detailed response. Below are my responses.
> > >
> > >
> > > Weakness 1: The benefits of using two separate policies.
> > > First, I think that lines 144-172 do not explain the benefits of using two separate policies clearly. Lines 144-155 introduce the procedure of LICRA. In lines 156-167, the authors claim that the decision space is reduced by using two separate policies. I think this claim is not true. Consider a simple approach that augments the action space with a null action and then performs any classical RL method. The decision space is $S(A+1)$. For LICRA, the total decision space is $S \times 2 + S \times A = S (A+2)$. Therefore, I think that the benefit that LICRA can reduce the decision space does not hold.
> > >
> > >
> > > Second, I think the experiments cannot validate the benefits of using two separate policies in LICRA. The mentioned simple baseline which augments the action space with a null action is not involved for comparison. Thus, I cannot see the benefits of additionally introducing two separate policies.
> > >
> > >
> > >
> > > Weakness 3: The clarity in section 5.
> > >
> > > In section 5, there are still some misleading parts. Therefore, it is difficult for me to evaluate the theoretical contribution here.
> > >
> > > First, the definition of $T$ in eq.(3) is problematic. The LHS term $T v^{\pi, g} (s_t)$ does not depend on any action $a$. However, the RHS term depends on some certain action $a$ through $\mathcal{M}^{\pi, g} Q^{\pi, g} (s_t, a)$.
> > >
> > > Second, according to the definition of $T$ in eq.(3), $T$ depends on some certain policy $(\pi, g)$ thorough $\mathcal{M}^{\pi, g}$. Therefore, in my understanding, $T$ is similar to the Bellman operator used in classical policy iteration, rather than the Bellman optimality operator in classical value iteration. If true, it is unreasonable that repeatedly applying the policy-dependent Bellman operator $T$ can converge to the optimal Q function.
> > >
> > > Third, in the Q-learning update rule in Theorem 2, they use the operator $M^{\pi, g}$ which depends on certain policies $(\pi, g)$. However, the part on how to choose $(\pi, g)$ is missing.

---

> > > > ### Author Response · Authors · 2022-08-06
> > > > **Response to the reviewer's comments**
> > > >
> > > > Thanks a lot for asking these important questions!
> > > >
> > > > **A key point.** A first thing to note is that in this setting since the agent faces a positive cost for each action, acting at each step is generally vastly suboptimal. In the case where the cost for acting is undiscounted, any policy that performs actions at each step would produce singular costs.
> > > >
> > > > **Point about the null action.** In our experiments, *all algorithms* apply actions that are drawn from an action set which contains a $0$ action. When this action is executed it produces $0$ costs and exerts no influence on the transition dynamics. This serves the role of a null action and is why we assert that our comparisons are valid.
> > > >
> > > > **Point about complexity.** We highlight the sequential aspect of LICRA means that the policy $\mathfrak{g}$ eliminates the need to execute the action-policy (and determine optimal actions) at every state but only the subset of states where an action is required. This set is in general smaller than the entire state space given our earlier remarks.  However, the reviewer’s calculation presupposes that an action is taken by LICRA’s action policy at each state. This is inconsistent with the setting we consider for the reasons mentioned above.
> > > >
> > > > **The reviewer’s complexity calculation.** We thank the reviewer for their time in detailing their thoughts. Given our remarks, using a similar line of reasoning as the reviewer, we can straightforwardly show that LICRA does indeed reduce complexity. We start by defining the set of states where the agent should act by $\mathcal{S}_I$ and its compliment by $\mathcal{S}_I^c$. With this, LICRA requires $|\mathcal{S}_I^c|+|\mathcal{S}_I||\mathcal{A}|$ calculations.
> > > > This is a reduction in complexity since $|\mathcal{S}_I^c|+|\mathcal{S}_I||\mathcal{A}|<|\mathcal{S}||\mathcal{A}|$ iff $|\mathcal{S}_I|(|\mathcal{A}|-1)-|\mathcal{S}|(|\mathcal{A}|-1)<0$, which is always true whenever $\mathcal{S}_I\subset \mathcal{S}$ which holds when $ |\mathcal{S}_I|<  |\mathcal{S}|$ i.e. it holds whenever the agent should not take an action at every state.
> > > > We did not include this reasoning as it removes focus from the very important case of continuous spaces.
> > > >
> > > > **Point about LICRA Q learning rule and $\mathcal{M}$.** The dependence of $\mathcal{M}^{\pi,\mathfrak{g}}Q$ on the action is eliminated by the fact that by definition (of $\mathcal{M}^{\pi,\mathfrak{g}}$) the action input is the action which is sampled from the policy $\pi$. This is analogous to the relationship between the value function $v$ and the action-value function $Q$ in standard reinforcement learning. In our Q learning rule, using the definition of $\mathcal{M}$ allows for any (admissible) policy $(\pi,\mathfrak{g})$ including epsilon greedy and or the Boltzmann distribution.
> > > > We again thank the reviewer for their suggestion on performing an analysis of our Q learning method. We have now uploaded a version of the paper which contains this analysis. Please see Section 12 of the supplementary material.
> > > >
> > > > **Point about LICRA's Bellman Operator.** Our calcuations include the term $Tv^{\pi, \mathfrak{g}}$ . The operator $T$ has no direct dependence on $(\pi, \mathfrak{g})$, rather the object that it acts on $v^{\pi, \mathfrak{g}}$ has this dependence (please see our earlier comment relating $v^{(\pi,\mathfrak{g})}$ and $\mathcal{M}^{(\pi,\mathfrak{g})}Q^{(\pi,\mathfrak{g})}$). In our latest upload of our work, we have now updated the notation in our definition of $T$ to avoid this confusion. This is analogous to the Bellman operator in standard RL acting on a value function $v^\pi$, the operand has a dependence on $\pi$ but the Bellman operator itself does not.
> > > >
> > > > We hope this completely clears everything up and are happy to answer anything that may remain unresolved!

---

> > > > > ### Comment · Reviewer_UN8M · 2022-08-08
> > > > > **Response to the authors**
> > > > >
> > > > > Thanks for the response. My question on the motivation for using two separate policies is addressed. However, I still have some concerns.
> > > > >
> > > > >
> > > > > **Questions on $\mathcal{M}^{\pi, g}$ and Bellman operator $T$.**
> > > > >
> > > > >
> > > > > According to your response, the term $\mathcal{M}^{\pi, g} Q^{\pi^\prime, g^\prime}$ should be defined as
> > > > >
> > > > > $\mathcal{M}^{\pi, \mathfrak{g}} Q^{\pi^{\prime}, \mathfrak{g}^{\prime}}\left(s_{\tau k}, a_{\tau k}\right) := \mathbb{E} _  { a_{\tau k} \sim \pi (\cdot| s_{\tau k}) }  [\mathcal{R}\left(s_{\tau k}, a_{\tau k}\right)-c\left(s_{\tau k}, a_{\tau k}\right)+\gamma \sum_{s^{\prime} \in \mathcal{S}} P\left(s^{\prime} ; a_{\tau k}, s\right) v^{\pi^{\prime}, \mathfrak{g}^{\prime}}\left(s^{\prime}\right) ]$
> > > > >
> > > > > In the revision, the expectation over $a_ {\tau k}$ is missing. However, the above definition is still problematic. The LHS term $\mathcal{M}^{\pi, \mathfrak{g}} Q^{\pi^{\prime}, \mathfrak{g}^{\prime}}\left(s_{\tau k}, a_{\tau k}\right)$ depends on certain action $ a_{\tau k}$ while the RHS term is independent of $a_{\tau k}$ due to the expectation.
> > > > >
> > > > >
> > > > > For the operator $T$, you said that the operand has a dependence on $\pi$ but the Bellman operator itself does not. If true, what is the definition of $Tv$ in Lemma 1? Here $v$ is not a value function of any policy.
> > > > >
> > > > >
> > > > > **Question on the Q-learning update rule.**
> > > > >
> > > > >
> > > > > My question is about the developed theory of Q-learning (i.e., Theorem 2). In the convergence analysis of Q-learning, how the policies $(\pi, g)$ are chosen? To my knowledge, without a smart exploration strategy, Q-learning cannot converge in polynomial time [1].
> > > > >
> > > > >
> > > > > Based on the above concerns, I encourage the authors to revise section 5 thoroughly. Be careful to introduce new definitions and notations, and replenish the missing definitions.
> > > > >
> > > > >
> > > > >
> > > > > [1] Chi Jin, Zeyuan Allen-Zhu, Sebastien Bubeck, and Michael I Jordan. Is q-learning provably efficient? In Advances in Neural Information Processing Systems, pages 4863–4873, 2018.

---

> > > > > > ### Author Response · Authors · 2022-08-08
> > > > > > **Response to the reviewer**
> > > > > >
> > > > > > We thank the reviewer for their comments.
> > > > > >
> > > > > > **Definition of $\mathcal{M}$.**
> > > > > >
> > > > > > In the definition of $\mathcal{M}^{\pi,\mathfrak{g}}$ the action $a$ is sampled from the policy in the term $\mathcal{M}^{\pi,\mathfrak{g}}Q(s,a)$ (c.f. line 194 of the definition of $\mathcal{M}$) i.e. $a\sim \pi$ and therefore the LHS of the expression is also not dependent on a choice of $a$ (analogous to the fact that $\max_yf(x,y)$ does not depend on a choice of $y$). Additionally as the action is *sampled* from the policy $\pi$ this definition does not introduce an expectation as the reviewer has expressed. For our convergence proofs of our LICRA Q-learning variant, the policy pair is restricted (please see our next comment).
> > > > > >
> > > > > > **The Bellman operator and our Q-learning rule**
> > > > > >
> > > > > > We thank the reviewer for raising this question. We agree with the reviewer’s point about the requirements for the convergence within RL. To explain our case, in general, the intervention operator $\mathcal{M}^{\pi,\mathfrak{g}}$ can have dependence on an arbitrary policy pair $(\pi,\mathfrak{g})$. For the case of our LICRA Q-learning variant, its convergence assumes that this policy pair is fixed as an epsilon-greedy policy (please see lines 553-554 in the supplementary material). This is carried into the Bellman operator and enables us to prove convergence of our Q-learning variant (in polynomial time). We thank the reviewer for raising this and will include a comment in our updated version so as to more clearly highlight this point.

---

> > > > > > > ### Comment · Reviewer_UN8M · 2022-08-09
> > > > > > > **Response to the authors**
> > > > > > >
> > > > > > > Thanks for the response. I think my concerns are not addressed.
> > > > > > >
> > > > > > > **Definition of $\mathcal{M}$**
> > > > > > >
> > > > > > > If you do not introduce an expectation in the definition of $\mathcal{M}^{\pi, g} Q (s, a)$, you cannot remove the dependence on a specific action $a$. Consider $\max_{y} f(x, y)$, you need the maximum operator to remove the dependence on $y$.
> > > > > > >
> > > > > > > **The Bellman Operator $T$**
> > > > > > >
> > > > > > > I think my question is misunderstood. I clarify my question as follows. In the definition of $T$ (line 202-203), you require that the operand has a dependence on $(\pi, g)$ (i.e., $v^{\pi, g}$). However, in Lemma 1, you apply $T$ on an arbitrary element $v$ in the vector space. What is the definition of $Tv$ if $v$ is not a value function of any policy?
> > > > > > >
> > > > > > >
> > > > > > >
> > > > > > > **Q-learning Convergence**
> > > > > > >
> > > > > > >
> > > > > > > I think that the argument that Q-learning with an epsilon-greedy policy can converge to the optimal policy in a polynomial time is not correct. In [1], they have shown that q-learning with an epsilon-greedy policy must require an exponential time complexity in some hard instances. Without a smart exploration strategy (e,g., upper confidence bound), we cannot expect that q-learning can converge to the optimal policy.
> > > > > > >
> > > > > > >
> > > > > > > [1] Chi Jin, Zeyuan Allen-Zhu, Sebastien Bubeck, and Michael I Jordan. Is q-learning provably efficient? In Advances in Neural Information Processing Systems, pages 4863–4873, 2018.

---

> > > > > > > > ### Author Response · Authors · 2022-08-09
> > > > > > > > **Response to the reviewer**
> > > > > > > >
> > > > > > > > We thank the reviewer again for their comments which we hope to now resolve.
> > > > > > > >
> > > > > > > > **Intervention operator $\mathcal{M}$**
> > > > > > > >
> > > > > > > > Our definition of $\mathcal{M}^{\pi,\mathfrak{g}}$ means that the second argument of the operand is sampled from the policy $\pi$ (and hence the variable is in fact a dummy variable). We could also load the notation and write $\mathcal{M}^{a\sim\pi,\mathfrak{g}}Q(s,a)$. To exemplify the point in a specific instance, consider a deterministic policy $f:\mathcal{S}\to\mathcal{A}$ and the term $\mathcal{M}^{f,\mathfrak{g}}Q(s,a)$. Then for this the second input to the $Q$ term is simply $f(s)$, i.e. $a\equiv f(s)$ and the whole expression is $\mathcal{M}^{f,\mathfrak{g}}Q^{\pi',\mathfrak{g}'}(s_\tau,a_\tau)=\mathcal{R}(s_\tau,f(s_\tau))-c(s_\tau,f(s_\tau))+\gamma\sum_{s'\in\mathcal{S}}P(s';f(s_\tau),s_\tau)v^{\pi',\mathfrak{g}'}(s')$.
> > > > > > > >
> > > > > > > > **The Bellman operator $T$ in Lemma 1**
> > > > > > > >
> > > > > > > > For our proof of Lemma 1, we fix the policy as an (epsilon-)greedy policy. This is given in line 554 where the action which is selected at the currect state is that which maximises [current reward - cost + discounted expected future return]. We also restrict the function $v$ to maps of the kind $v:\mathcal{S}\to\mathbb{R}$ (c.f. lines 549-550) therefore $v$ in this instance can be interpreted as a value function of an (epsilon-)greedy policy.
> > > > > > > > Concretely, from line 554, the definition of $Tv$ here is:
> > > > > > > > $Tv(s_{\tau}):=\max\Big\\{\max_{a\in\mathcal{A}}\\{\mathcal{R}(s_{\tau},a)-c(s_{\tau},a)+\gamma\sum_{s'\in\mathcal{S}}P(s';a,s_{\tau})v(s')\\},
> > > > > > > > \[\mathcal{R}(s_{\tau},0)+\gamma\sum_{s'\in\mathcal{S}}P(s';0,s_{\tau})v(s')]\\}$
> > > > > > > >
> > > > > > > >
> > > > > > > >
> > > > > > > > **Our convergence proof**
> > > > > > > >
> > > > > > > > We thank the reviewer for this question. Our paper proves convergence of the *value function* $Q$ to the optimal value function $Q^\star$. As with standard Q learning, this is provable under the standard assumptions of stochastic approximation theory, e.g. [2]. This does not assert claims of polynomial-time learning of the *optimal policy* (in episodic settings which is the task and problem setting being considered in [1].)
> > > > > > > >
> > > > > > > > [2] Tsitsiklis, J. N. (1994). Asynchronous stochastic approximation and Q-learning. Machine learning, 16(3), 185-202.

---

> > > > > > > > > ### Author Response · Authors · 2022-08-09
> > > > > > > > > **Comment to the reviewer from the authors**
> > > > > > > > >
> > > > > > > > > We great appreciate the reviewer’s effort in providing suggestions to improve the paper and raising questions for us to address their concerns. We would like to ask the reviewer if they have any issues with the paper. If none remain, we would be grateful if the reviewer could adjust their score to reflect their most current appraisal of our work (if they see fit to change it from reject).

---

> > > > > > > > > ### Comment · Reviewer_UN8M · 2022-08-10
> > > > > > > > > **Response to the authors**
> > > > > > > > >
> > > > > > > > > Thanks for the clarification. I will increase my score to 5. Below are some suggestions.
> > > > > > > > >
> > > > > > > > > To formalize the meaning that the second argument of the operand is sampled from the policy, I think you should introduce an expectation operator.
> > > > > > > > > $
> > > > > > > > > \mathcal{M}^{\pi, g} Q^{\pi^\prime, g^\prime} (s_\tau) = \mathbb{E} _ {a_{\tau k} \sim \pi\left(\cdot \mid s_{\tau k}\right)}\left[\mathcal{R}\left(s_{\tau k}, a_{\tau k}\right)-c\left(s_{\tau k}, a_{\tau k}\right)+\gamma \sum_{s^{\prime} \in \mathcal{S}} P\left(s^{\prime} ; a_{\tau k}, s\right) v^{\pi^{\prime}, \mathfrak{g}^{\prime}}\left(s^{\prime}\right)\right]
> > > > > > > > > $
> > > > > > > > >
> > > > > > > > > For the definition of $Tv$ and the convergence proof, you can replenish these necessary definitions and explanations to the main text.

---

> > > > > > > > > > ### Author Response · Authors · 2022-08-10
> > > > > > > > > > **Response to the reviewer**
> > > > > > > > > >
> > > > > > > > > > We thank the reviewer for their recent suggestion and oveall comments and lively discussions that have enabled us to greatly improve the paper (please see the very last upload). We are happy to add more finishing adjustments in the camera-ready should the paper be accepted.
> > > > > > > > > >
> > > > > > > > > > We noted that all reviewers have agreed in full on the importance of the problem setting we have tackled that is currently unaddressed and would be of benefit to the RL community. Our paper, which is supported by elaborate theory and empirical evalutions, now contains a very comprehensive analyses of our novel method that provides some deep insights about LICRA. This includes empirical analyses to validate our theoretical claims, empirical evaluations in 3 domains and 3 ablation studies.
> > > > > > > > > >
> > > > > > > > > > We along with the reviewer have also expended a great deal of effort to ensure confidence in the correctness of our claims, theory and given detailed justifications for our method. We now also believe we have resolved any concerns from all reviewers and delivered on all the reviewers' many and diverse requests.
> > > > > > > > > >
> > > > > > > > > > While we appreciate the reviewer changing their score, given the reviewer's current score of a borderline accept is reserved for special cases where the contribution is only very slight, we respectfully ask the reviewer to reconsider the modest movement and consider a more substantial upward adjustment.
> > > > > > > > > >
> > > > > > > > > > Best wishes,
> > > > > > > > > >
> > > > > > > > > > The authors

---

### Official Review · Reviewer_A4kA · 2022-07-11

**Rating:** 6
**Confidence:** 3
**Soundness:** 3 good
**Presentation:** 2 fair
**Contribution:** 2 fair

**Summary:**

This paper proposes an algorithm for finding the optimal policy in sequential decision problems where there is a cost associated with taking an action. At each timestep, the agent can choose to perform an action or do nothing. LICRA learns two RL policies: one for deciding when to act, and another for taking actions. By first learning when to act, the complexity of the problem is reduced compared to standard RL configurations.

**Questions:**

* In Figure 1, what are the measures of centrality and spread for the different algorithms? How many random seeds were used? How much hyperparameter tuning was necessary to achieve the results shown. Same questions for Figures 2 and 3.
* Please provide brief descriptions of each baseline algorithm used, and why they were chosen to be compared with LICRA.
* What do minibatches refer to in Appendix 12?
* What do optimization epochs refer to in Appendix 12?
* In Appendix 12, there are no hyperparameters in square brackets following the notice in the second sentence. This section seems very incomplete to me. The SAC and PPO variants of LICRA should have different sets of hyperparameters. Are these hyperparameters also used for the baseline algorithms? Are these hyperparameters the same for both $\pi$ and $g$?
* Are the costs included in the Average Episodic Return reported in the figures? It could be insightful to compare the baselines using the vanilla reward.

**Limitations:**

I think the potential limitations of LICRA should be discussed in more detail. The experimental environments chosen are very well suited to LICRA since they involve the agent positioning itself in a certain configuration (e.g., spaceship over the goal in lunar lander) and maintaining that configuration by not acting. It would be nice to see how LICRA performs in environments that are less suited to it. For example, what about tasks where the agent should be acting “most” of the time, making it harder for $g$ to learn the few states where it should intervene?

**Strengths And Weaknesses:**

#### Originality
The authors present a novel method for impulse control using two RL policies.

#### Quality
The paper does a nice job of showcasing practical applications of LICRA in environments such as finance and autonomous driving. I did not examine the proofs in the appendix very closely, but the results seem to follow.

I think the related work section is lacking descriptions and citations for related problem settings such as RL safety and hierarchical RL. Many of these works use a safety policy or high-level policy to intervene the lower-level policy during training. Since LICRA trains two RL processes, it would be interesting to see how it compares to other methods that do the same.

#### Clarity
The paper is well structured and easy to understand. Algorithm 1 presents a procedure that should be easy to implement with standard auto-differentiation libraries. During the exposition of LICRA, the authors state that adding the zero action is not a good solution to impulse control. I think a small motivational example could be good for demonstrating this.

##### Nits
* The last paragraph of the introduction should be rewritten for grammar and clarity
* Line 28: result "in" catastrophic losses
* Line 56: series of results
* Line 82: form of
* Line 90: task is to
* Line 135: not significantly impact
* Line 190: $P(\cdot ;  a_{\tau_k}, s_{\tau_k})$

#### Significance
The results are useful for applications involving impulse control. Researchers or practitioners may use LICRA due to its effectiveness and how easy it is to implement in practice.

---

> ### Author Response · Authors · 2022-08-02
> **Response to the reviewer A4kA**
>
> We thank the reviewer for their reading and many insightful comments. We are glad the reviewer appreciates the clarity of the paper and the plug \& play aspect which allows for quick implementation. We address the reviewer's comments and questions individually.

---

> ### Author Response · Authors · 2022-08-02
> **Response to the reviewer A4kA**
>
> > Measures of centrality and spread, number of seeds
>
> #### Answer: The line indicates mean performance and the shaded area is one standard deviation. For Merton problem, all algorithms were trained 5 times for 5 different seeds. For Drive environment problem, all algorithms were trained 3 times for 3 different seeds. For LunarLander environment, all algorithms were trained 5 times for 5 different seeds. We apologise for not displaying this clearly and will update the script accordingly.
>
> > Brief descriptions of each baseline algorithm used, and why they were chosen
>
> #### Answer: Thanks for the suggestion. We will now update the paper to include these details.
>
> > "Minibatches" and "optimization epochs" in Appendix 12?
>
> #### Answer: Apologies for this oversight. Minibatches mean the sampled batches of experince for mini-batch learning process. Epochs mean the times of PPO updates in a paramter optimization process (not necessary to set for SAC).
>
> > Missing information on hyperparameters
>
> #### Answer: We apologise for this oversight. For LunarLander experiment, PPO and SAC use the same set of hyperparameters while PPO should consider some extra hyperameters (i.e., number of PPO epoches, whether to use GAE). These hyperparameters are also used for the baseline algorithms, which are the same for both $\pi$ and $g$.
>
> > Are the costs included in the Average Episodic Return reported in the figures?
>
> #### Answer:
> Yes, the average episodic return includes both the reward and cost.

---

> > ### Comment · Reviewer_A4kA · 2022-08-07
> > **Thanks for your response**
> >
> > >>Brief descriptions of each baseline algorithm used, and why they were chosen
> > >Answer: Thanks for the suggestion. We will now update the paper to include these details.
> >
> > Where can I find these in the updated paper?

---

### Official Review · Reviewer_wRMk · 2022-07-26

**Rating:** 5
**Confidence:** 2
**Soundness:** 2 fair
**Presentation:** 2 fair
**Contribution:** 3 good

**Summary:**

According to the authors, this paper studies the problem of choosing when to act, also known as the impulse control problem. Choosing to abstain from actions has relevance in various real-world scenarios where doing nothing can be the right decision. As the authors discuss, this can be because of some costs associated with taking action. A motivating example of their work is portfolio management, where not buying or selling financial instruments is the right move as fees are associated with each action.

The authors model propose a framework that incorporates the impulse control problem in the reinforcement learning problem. They accomplish this by representing costs associated with actions taken in the reward function. They decompose the agent policy into two policies: the original action space policy and a binary policy to choose whether to act or not. They argue against including this action in the rest of the action set because it would be inefficient in the learning process due to action cardinality.

The authors further prove theoretical results that their approach will converge to the optimal Q function. They do this in the tabular and linear function approximator cases. Their experiments further validate the framework, and the authors suggest their framework appropriately deals with these challenges empirically compared to baselines.


**Questions:**

- Are features shared for both policies in the experiments or are they entirely separate?
- Can the action decision policy be a separate linear head trained with a different reward signal?
-  Were any experiments done to validate the theoretical results?
- A statement in section 4 seems important to evaluate: "if the ratio of s to s_i is large, learning g before pi is essential". What was the rationale for abstaining from testing this scenario?
- In theorem 2, the max is between taking action or future expected reward after doing nothing. In terms of having taken action, why is it a_t and not max_{a_t}? Wouldnt this make this make it more of a SARSA update?
- What is delta in section 6 Equation 4? I could not find the definition.
- What is the state variable of the Merton problem? Does it have any sequential relations?
- In the Merton experiments, are the results sensitive to the environment parameters of the SDE? I.e., is LICRA robust to these choices?
- Which hyperparameters were tuned over? It is mentioned in the 12 Hyperparameter section that "square brackets indicate ranges," but I did not see any in the hyperparameter in the table using square brackets.
- Equation 14 of proof of theorem 1, inconsistent usage of lambda (is its function of 1 or 2 inputs?). What does "I(\tau)" mean in Equation 14?
- Please tighten up notation in theorem one proof; the proof starts with \lambda, which becomes \phi and then \theta. It is not unclear why these variables change.
- A paper that might be related is the following: Nonparametric learning for impulse control problems Soren Christensen and Claudia Straugh. Disclaimer: I thoroughly read the paper, but they mention impulse control and reinforcement learning
- Would it be possible to include in the preliminary discussion on Impulse Control Problem? This seems central to your research but is omitted in the paper.
- Consider restructuring your existing experiments section. It feels like Figures 1, 2, and 3 are, in some sense, experiments of similar conclusions (i.e., our method performs well). The ablations you do in the appendix seem valuable to include in the main paper as it isolates the effects of hyperparameters of the framework.
- Consider moving the less crucial details of the Merton problem to the appendix. For example, the SDE feels unnecessary in the main paper.
- You reference Fig. 5 in line 462 when it seems you mean Figure 4.
- Consider removing the examples in "3 Preliminaries". They do not add much to the discussion that was not already mentioned in the introduction.
- Consider using a more digestible metric for performance over learning curves (e.g. Final average performance). It’s a bit difficult to distinguish the performance gains of your algorithm.
- Include Compute resources used. This is mentioned as required in the Checklist, but these details weren’t mentioned in the appendix or paper.
- Figure 6: If you change to a single metric (average final reward, the area under the curve, success rate, etc.) You could consider using a radar plot to visualize performance. As higher is assumed to be better, you could have the rewards on the outside, and each radar is your algorithm. The bigger the circle, the better your algorithm is across cost functions.
- What is Figure 5 showing exactly? As K goes up, fewer violations should occur as it's terrible to decelerate in these zones, but it wasn't clear. It could help to visualize this experiment to explain it.


**Limitations:**

It's hard to gauge the limitations of their work because the authors do not seem to discuss this explicitly in the paper. In terms of negative societal impact, there doesn't seem to be any apparent reason their work could have negative social impacts.

**Strengths And Weaknesses:**

According to the authors, the paper presents a significant problem that has not been studied in the reinforcement learning problem. From the reviewer's experience in robotics, it is a valid consideration that not sending a new action can be the optimal decision. The author's theoretical results support the validity of their proposed algorithms. The fact that their algorithm can augment existing reinforcement learning algorithms is a great advantage as it means one can quickly implement the additional policy for real-world applications. The way the authors structured their problems seems appropriate to understand the problem they are interested in addressing.


However, a concern with this paper is with the experiment evaluations. Much more evaluation is necessary to validate the LICRA framework. A few crucial things that come to mind would be to validate the theoretical results in the tabular and linear cases. The authors discuss Q-learning-like algorithms, but it seems like they do not even propose it by itself as a new algorithm or do any experiments with it. Not evaluating this update rule feels like a missed opportunity to isolate and understand the influence of the LICRA framework in more easily reproducible settings.

Supposing the there were reasonable justification against these types of experiments, their existing experiments could still be improved. One discrepancy in the experiments is the different versions of LICRA algorithms that the authors claim to study instead of report. In the Experiments introduction, they introduce LICRA_PPO and LICRA_SAC, but in Figure 1, LICRA_SAC is not present; in Figure 2, there is just "LICRA,"; and only in Figure 3 are both of these algorithms present. Perhaps this is nit-picking, but from the author's inclusion of different combinations of RL policies with their algorithm, one would expect full coverage of the combinations (so four different versions of LICRA if using both SAC and PPO for each sub-policy). Additional comments are left for the question section as they are more about details to be included and for clarification.

---

> ### Author Response · Authors · 2022-08-02
> **Response to the reviewer wRMk**
>
> We thank the reviewer for their reading and many insightful comments. We are glad the reviewer appreciates the importance of the problem which at present is unstudied within the context of learning and RL, and the plug \& play aspect which allows for quick implementation. We address the reviewer's comments and questions individually.

---

> ### Author Response · Authors · 2022-08-02
> **Response to the reviewer**
>
> >"More evaluation is necessary" and Q-Learning like algorithms.
>
> **Answer**
>
> We thank the reviewer for their insightful suggestions which have now improved our paper. We firstly remark that LICRA is a general framework which can accommodate existing RL algorithms to enable them to efficiently solve problems studied in our paper. To exemplify this point, in our experiments we considered variants of the LICRA method showing it seamlessly adopts existing methods.
>
> For the case of the LICRA Q-learning variant, Theorem 2 proves its convergence. As such this also serves as the foundation for establishing results for actor-critic methods which extend value-based learning to included policy updates. We have now included a comprehensive empirical analysis of the LICRA Q-learning variant of Sec. 4 showing the convergence of the Q function. We consider the tabular case showing convergence consistency with Theorem 3.
>
> >Different variants of LICRA
>
> **Answer**
>
> We again thank the reviewer for this comment. We will update our experiments to include different versions of LICRA in the intervention policy (please see the updated script). We will include full coverage of the 4 combinations in our script in our next update.
>
> > Are features shared for both policies in the experiments or are they entirely separate?
>
>
> **Answer** The features in our experiments are not shared between policies, however, in principle it would be possible to use LICRA where features are shared.
>
> > Can the action decision policy be a separate linear head trained with a different reward signal?
>
> **Answer** Thanks for the interesting suggestion. It is possible to train a policy with different heads for $\pi$ and $\mathfrak{g}$ to improve efficiency. We have not yet tried this but we are keen to see if it delivers improvements.
>
> > Were any experiments done to validate the theoretical results?
>
> **Answer**  Yes, please see our earlier comment. We now add  a new section in the Appendix in which we empirically validate the results of Theorems 2 and 3.
>
> > LICRA's increased efficiency given a large difference between $\cal S$ and the subregion in which it is optimal to intervene (intervention region) $\cal S_i$
>
> **Answer**
>
> We thank the reviewer for the great suggestion. We will now include a section in the Appendix in which we perform an empirical analysis where we indeed validate this claim showing that LICRA is extremely efficient at learning the optimal policy in a modified version of Drive env. where the optimal policy requires the agent to act only at a very subregion of the state space
>
> > The Max operator in the algorithm update in Theorem 2.
>
> **Answer**
> The current action (if executed due to the decision by $\mathfrak{g}$) is sampled from the current policy $\pi$ as per the definition of $\mathcal{M}^{\pi,\mathfrak{g}}$; the max operator (over the agent's action space) affects only the expected future return . This is analogous to the update rule of standard Q learning see for example equation 1 in [1].
>
> [1] Even-Dar, Eyal, Yishay Mansour, and Peter Bartlett. "Learning Rates for Q-learning." JMLR (2003).
>
> > The delta in Equation 4?
>
> **Answer** Apologies for this oversight. The delta function is the Kroncker-delta function which is now defined on line 241. We have highlighted this in the current version for the reviewer's convenience during the rebuttal phase where we added the following definition:
>
> $$\sum_{k\geq 1}\delta^t_{\tau_k}=\begin{cases} 1, \text{if an impulse was applied at time }  t\\\\0, \text{otherwise }\end{cases} $$
>
> While this definition is somewhat awkward to read, it is consistent with the impulse control literature.

---

> > ### Author Response · Authors · 2022-08-03
> > **Response to Reviewer wRMk continued...**
> >
> > > State variable of the Merton problem
> >
> > **Answer**  The state variable is $W_t$ (the investor's wealth) whose transition dynamics are described by equation (7). We have now made a brief comment in the text to highlight this.
> >
> > > Details of the hyperparameters
> >
> > **Answer**  Apologies for this oversight. For these mentioned hyperparameters, we tune Rollout length within $[32,64,128,256]$. We will update the paper to include this in our revised version.
> >
> > > Lambda and "I(\tau)"  and notation in the Theorem 1 proof.
> >
> > **Answer**
> >
> > Thanks for pointing this out (each are just functions in $L_2$). We have made this notation consistent in our updated version. We removed $I(\tau)$ as this was a typo.
> >
> > > The paper "Nonparametric learning for impulse control problems" by S. Christensen, C. Straugh
> >
> > **Answer**  Thanks a lot for this reference. This paper is only loosely related to our work (beyond the impulse control we already discussed), since they consider a continuous-time setting where they assume the reward function is known and knowledge of the probabilitistic component of the dynamics (specifically the coefficient of the noise term). Their analysis uses a data-driven approach namely, parametric estimation for the value function and opposed to reinforcement learning (RL). This is markedly different to our RL approach which is supported by an implementable algorithm and enables us to solve completely unknown settings. We will now include a brief statement in relation to this work.
> >
> > > Inclusion of discussion on the Impulse control
> >
> > **Answer** We refer the reviewer to lines 81-88 where we survey the impulse control literature. Note that the literature is vastly dominated with continuous-time analyses (within known environments) which leads to an (optimal stochastic control) approach the markedly differs from ours.
> >
> > > Restructuring the experiments section.
> >
> > **Answer** Thanks for the very valuable suggestion. We also agree that the Ablation Study adds a valuable analysis which deserves to be in the main body. We have now adjusted it in the new update.
> >
> > > More digestible metric for performance.
> >
> > **Answer**  Thanks for the suggestion. We have now added the following final scores to the captions that clearly indicate LICRA's superior empirical performance.
> > * LunarLander(final scores):
> > LICRA_SAC(98.03), SAC(61.20), LICRA_PPO(58.84), PPO(49.77).
> > * Drive Env(final scores):
> > LICRA_SAC(1.75), SAC(0.37), LICRA_PPO(2.63), PPO(0.24)
> >
> > >Compute resources used.
> >
> > **Answer** Thank you for pointing out our oversight. LICRA does not require large resources to perform the experimentation in the paper. The most computationally intensive experiments can be run using a laptop machine with 8GB memory. We will update the script accordingly with all these details.
> >
> > > What Figure 5 shows
> >
> > **Answer** The analysis of Fig 5. shows that LICRA learns to prioritise actions at states where not acting produces the highest penalties. The lower region of the map incurs the highest penalties for moving at a velocity less than $v_{min}$ whereas the highest part incurs low costs. As the cost for acting increases (i.e. as we make $\\{c_k\\}$ larger), LICRA learns it should expend costs for acting in the lower region while ''`taking the hit'' in the higher regions.

---

> ### Author Response · Authors · 2022-08-08
> **Did we address your concerns?**
>
> Dear reviewer wRMk,
>
> We are grateful for your review of our paper.
>
> We have sought to address each of your points in detail during our rebuttal. We believe all the points you have raised have been entirely addressed during the course of our rebuttal.
>
> We have not received a response from you during the rebuttal period.
>
> We would be grateful if you could confirm our responses have resolved all of your comments or let us know if any points of yours remain unaddressed.
>
> Best wishes,
>
> The authors

---

> > ### Author Response · Authors · 2022-08-08
> > **Further updates**
> >
> > Further to our last comment.
> >
> > We have also uploaded and update to our paper which now includes an additional section in the supplementary material as well as a more comprehensive set of baselines within our experiments. We have elaborated the details of the updates in our note to all reviewers.
> >
> > Best wishes,
> >
> > The authors

---

> > > ### Comment · Reviewer_wRMk · 2022-08-09
> > > **Additional questions based on the updates.**
> > >
> > > First and foremost, thank you for taking the time to incorporate the feedback into the paper. Thank you for pressing for additional feedback. What follows are thoughts from reviewing your comments and portions of the recently updated draft:
> > >
> > > - Thank you for adding the experiments validating Theorem 2 & 3 in the appendix. Remember to reference this in the main paper. More importantly, What is the explanation for the massive variance in Figure 8? How many random seeds were used in these experiments? Could you provide additional details on this testing bed used to validate these results? It sounds like a sort of bandit problem, but it could be challenging to reproduce the plots without knowing the exact details of how you experimented.
> > >
> > > - As a follow-up to the above, an important series of experiments that seems to be missing are validations of the linear function approximation case. Have you conducted any experiments in this scenario? Considering the Merton-Portfolio problem is one-dimensional, it seems to be the perfect problem to validate LICRA in the linear approximation setting.
> > >
> > > - Some other linear experimental test beds one might consider might be mountain cars or cart-pole. One could modify them similarly where the goal is to minimize the necessary amount of input control.
> > > Another thought for abstaining from an action might be some Roulette world where there are many betting actions. Using the LICRA policy could (I hypothesize) help speed convergence to choosing not to play the game entirely sooner then including it as another action. This setting could focus on the value of separating this decision from a more extensive action set.
> > >
> > > - Figure 1 It seems like the performance gains of LICRA PPO are only early in training, but asymptotically the two policies converge to the same mean reward. How do you explain this observation?
> > >
> > > - Thank you for clarifying my confusion about whether there should be a max operator or not in Theorem 2
> > >
> > > - Thank you for clarifying the state representation of the Merton problem.
> > >
> > > - Please include all algorithms' error bars for the reported final score. For example, LunarLander (final scores): Licra_SAC(98.03), SAC(61.20) …; what are the error bars for these and other experiments?
> > >
> > > - Another comment noted that only up to 5 experiments were run depending on the environment. What was the rationale for not conducting more repetitions of the experiment?
> > >
> > > - Thank you for clarifying Figure 5. Based on your clarification, I understand that LICRA learns to take more actions when the penalty for in-action is worse than the cost incurred for acting. Does this sound correct?
> > >
> > > - Thank you for including the discussion on tuning roll-out length. However, often deep RL can be highly dependent on seemingly innocuous hyperparameter choices. Given this potential challenge, what were the justifications for not doing a more extensive sweep? One idea to limit resources might be a random search over defined prior distributions for several hyperparameters, limiting the max number of configurations you do and then repeating each model.
> > >
> > > - Knowing the resources you used to conduct all experiments is the more important detail for the Compute resources comment. For instance, was a compute cluster used, what types of GPUs were used (if any), and how many CPUs were available?
> > >
> > >
> > > Minor comments:
> > >
> > > - If you want to keep the examples in “3 Preliminaries,” make sure to tighten the paragraph headers. For the first example, you name it and even have a citation, but for “Example 2” (line 137) you don’t mention autonomous driving in the header.
> > >
> > > - Line 202 - 209: When you describe each component in Equation 3, consider including the equation portion you are referring to in the sentence. For example, “the first term M^{\pi, g} … is the expected ….”.
> > > - Line 102: A suggestion previously I’ve heard when directly referring to a citation is to use <Last name> et al. year (or just citation), so instead of In [26], put In Pang et al. [26] or Pang et al. 2021…
> > >
> > > - Fix the labeling of algorithms in plots; for example, in Figure 7 of the Appendix, you have “Licra_PPO” and “Licra_SAC.”. Consider changing these to “Licra PPO” and “Licra SAC”. This also appears in Figure 1 (Licra_ppo). Also, in the main text (starting at line 271), consider renaming LICRA_PPO to something like LICRA-PPO or LICRA-PPO.

---

> > > > ### Author Response · Authors · 2022-08-09
> > > > **Response to the reviewer**
> > > >
> > > > Thanks a lot to the reviewer for getting back to us and providing great feedback.
> > > >
> > > > **Additional details for the Q-learning analysis**
> > > >
> > > > Thanks for raising the points of the extra details of the analysis we have just included. We will include the full details the reviewer has highlighted in our updated version. Briefly, the experiment setting was a simple 2-dimensional “gridworld” setting with a continuous action space and target goal state. We agree with the reviewer about testing the linear function approximation case, in fact this experiment tested exactly that. We find the reviewer’s thoughts on other environment settings very interesting and are grateful for the suggestions. We will consider adding an additional experiment though we are mindful of the fact that with the additional analyses the paper is already very comprehensive and lengthy.
> > > >
> > > > **Asymptotic convergence of LICRA\_PPO vs PPO**
> > > >
> > > > Our framework induces *efficient* learning in problems where the agent faces a fixed positive cost for performing actions. We expect asymptotically, in principle algorithms such as PPO can eventually find the same high performance solutions as LICRA but will require a great number more samples – this is what we observe in Fig. 1 where the problem setting is very simple. In practice, and given more complex problem settings, we observe that given a reasonable training budget PPO and other baselines do not learn the high performance policies as learned by LICRA (and may fail to solve the task altogether) – this is the case for all our other experiments.
> > > >
> > > > **Including other details**
> > > >
> > > > We are very grateful to the reviewer for their suggestions about including other important details. As during the rebuttal, we are committed to augmenting the paper to be as comprehensive as possible in this area – we will therefore update the camera ready paper with such details should the paper be accepted.
> > > >
> > > > **Reviewer’s statement about Fig. 5.**
> > > >
> > > > The reviewer’s understanding of LICRA’s behaviour in figure 5 is absolutely correct. We find this behaviour both useful and intuitive. We believe this is one of the many interesting features that our framework exhibits when augmenting standard RL methods.
> > > >
> > > > **Other helpful comments to improve the paper.**
> > > >
> > > > We greatly appreciate the reviewer’s effort in providing suggestions to improve the paper. We are performing these updates as we speak. We would like to ask the reviewer if they have we can address any concerns that may remain. If we have resolved all of the reviewer's concerns, we would be grateful if the reviewer could adjust their score to reflect their most current appraisal of our work (if they see fit to change it from reject).

---

> > > > > ### Comment · Reviewer_wRMk · 2022-08-09
> > > > > **Further clarification on decision when choosing hyperparameters**
> > > > >
> > > > > I think the primary concerns I have remaining are just around the experimental evaluation set-ups. It seems the comments which focus on justifying decisions around hyperparameter choices or the number of experiment evaluations have gone unaddressed. Knowing why a more rigorous evaluation of the potential hyper-parameters in the deep learning algorithms was not considered would be appreciated. As previously mentioned, deep RL  can be quite dependent on hyperparameters, so without extensive parameter sweeps the robustness of your conclusions is debatable.

---

> > > > > > ### Author Response · Authors · 2022-08-09
> > > > > > **Response to the reviewer**
> > > > > >
> > > > > > We thank the reviewer for their comments.
> > > > > >
> > > > > > LICRA is a plug and play framework which accommodates existing RL algorithms enabling them to tackle the current problem setting. In our paper we have used popular RL algorithms as our “base learners” for LICRA. As such tuning the hyperparameters for LICRA involves tuning over the same hyperparameter spaces as the baselines we compared to - indeed the hyperparameters of our baselines were sweeped over comprehensively.
> > > > > >
> > > > > > We are happy to include all such details of this in the camera-ready version should the paper be accepted.
> > > > > >
> > > > > > In terms of numbers of evaluations, for the Merton problem, all algorithms were trained 5 times for 5 different seeds. For Drive environment problem, all algorithms were trained 3 times for 3 different seeds. For LunarLander environment, all algorithms were trained 5 times for 5 different seeds.
> > > > > >
> > > > > > We have now included all such details in the paper.

---

### Author Response · Authors · 2022-08-02
**Note to all Reviewers**

We would like to thank the reviewers for their careful reading and insights that have improved the paper with more comprehensive analyses. Additionally, we thank the reviewers who have expressed their appreciation of the importance of the problem we tackle and the clarity of our exposition.

The purpose of the paper is to introduce a framework within reinforcement learning that easily incorporates existing methods enabling them to efficiently solve problems in which the controller faces costs for taking actions. Our experiments verify our claim that such technology improves RL performance and enables RL methods to consistently solve problems of this kind where existing methods perform badly and may sometimes even fail.


**We outline some of the items we have now added to the manuscript to complement the already in-depth analyses we have performed in the paper:**

* **Comprehensive empirical analysis to validate the convergence theorems of LICRA Q learning variant (Theorem 2 and Theorem 3) (see Section 12 of the supplementary material)**
* **New variants of LICRA to fully demonstrate LICRA's plug \& play capabilities.**
* **Suite of new baselines to verify the claim that LICRA outperforms current state-of-the-art RL methods.**
* **Additional details relating to experimental settings.**

These analyses now support the suite of ablation studies that yield insights into the behaviour of the policies LICRA generates (see Ablation study 1) and LICRA's behaviour when faced with different forms of costs (see Ablation study 2).

We also restructured the experiment section in order to bring the insights gained in the ablation studies into the main body.

In our updated version, we will also add:

* **Experiments to validate the claim that when the subregion of the state space in which the agent is required to act (intervention region) is relatively small, LICRA's approach vastly improves efficiency.**

**EDIT: This has now been added - please see Sec. 11.2 of the supplementary material.**

We hope the reviewers appreciate the updates to our paper.

**EDIT: We believe we have now resolved **all** the reviewers' questions including the very in-depth inspection of our theoretical analyses (see the points raised and inspection by reviewer UN8M) and each of the claims made about our framework each of which are now supported by empirical results some of which were added at the request of the reviewers.**

---

### Meta-Review · Area_Chair_dV8S · 2022-08-30

**Recommendation:** Reject
**Confidence:** Less certain

**Metareview:**

After careful consideration, I feel I must reject this paper.  This paper essentially proposes a specific hierarchical structure onto the action space for a specific set of MDPs (which motivated the paper), where all actions apart from one (the 'do nothing' action) incur some cost.

The proposed mechanism is a reasonable solution technique for such problems.  But the fact that hierarchical methods for specific problems that inject a form of domain knowledge are somewhat better than methods that do not has been shown many times, and is unsurprising.

The proposed method is, for instance, brittle to slight variations of the problem: what if some actions sometimes do or do not incur costs, for instance, rather than there being one clear such action? If you do nothing to a robot and it then falls off the table, does or does that not count as an immediate cost? Also, the framing of the paper excludes the 'do nothing' action from the 'original' action space, which - as discussed with reviewers - is a strange framing: why was a valid action, which apparently is often the optimal choice, excluded from the original action space?  And, more importantly, what if we just add it back?

I found the explanations of the authors for why to use the proposed mechanism rather than considering the flat action space to be ultimately unconvincing and, more importantly, rather domain specific and brittle.  One could argue that the proposed mechanism is sometimes a useful inductive bias, for some problems, which is not very surprising.  Other action decompositions will be useful for other problems.

Ultimately, I think the paper should 1) make a better case for why and when this particular hierarchy is important and beneficial compared to just allowing the system to decide for itself when to take each action (including the 'do nothing' action), 2) should be more upfront about limitations (e.g., when is this particular inductive bias a bad idea, rather than beneficial?), and 3) position itself better in existing work (e.g., show more awareness about prior hierarchical approaches - I would argue this is not really a 'novel method', but instead a specific instance of a hierarchical RL system).

I will therefore suggest to reject the paper so that the authors have an opportunity to use these comments to improve the paper.

**Award:**

No

---

### Decision · Program_Chairs · 2022-09-14

Reject